# TRAINING PROVABLY ROBUST MODELS
# BY POLYHEDRAL ENVELOPE REGULARIZATION

## ABSTRACT

Training certifiable neural networks enables one to obtain models with robustness guarantees against adversarial attacks. In this work, we use a linear approximation to bound model's output given an input adversarial budget. This allows us to bound the adversary-free region in the data neighborhood by a polyhedral envelope and yields finer-grained certified robustness than existing methods. We further exploit this certifier to introduce a framework called polyhedral envelope regularization (PER), which encourages larger polyhedral envelopes and thus improves the provable robustness of the models. We demonstrate the flexibility and effectiveness of our framework on standard benchmarks; it applies to networks with general activation functions and obtains comparable or better robustness guarantees than state-of-the-art methods, with very little cost in clean accuracy, i.e., without over-regularizing the model.

## 1 INTRODUCTION

Despite their great success in many applications, modern deep learning models are vulnerable to adversarial attacks: small but well-designed perturbations can make the state-of-the-art models predict wrong labels with very high confidence (Goodfellow et al., 2014; Moosavi-Dezfooli et al., 2017; Szegedy et al., 2013). The existence of such adversarial examples indicates unsatisfactory properties of the deep learning models' decision boundary (He et al., 2018), and poses a threat to the reliability of modern machine learning systems.

As a consequence, studying the robustness of deep learning has attracted growing attention, from the perspective of both attack and defense strategies. Popular attack algorithms, such as the fast gradient method (FGM) (Goodfellow et al., 2014), the CW attack (Carlini & Wagner, 2017) and the projected gradient descent (PGD) algorithm (Madry et al., 2017), typically exploit the gradient of the loss w.r.t. the input to generate adversarial examples. To counteract such attacks, many defense algorithms have been proposed (Buckman et al., 2018; Dhillon et al., 2018; Guo et al., 2017; Ma et al., 2018; Samangouei et al., 2018; Song et al., 2017; Xie et al., 2017). However, it was shown in Athalye et al. (2018) that most of them depend on obfuscated gradients for perceived robustness. In other words, these methods train models to fool gradient-based attacks but do not achieve true robustness. As a consequence, they become ineffective when subjected to stronger attacks. The only exception is adversarial training (Madry et al., 2017), which augments the training data with adversarial examples. Nevertheless, while adversarial training yields good empirical performance, it still provides no *guarantees* of a model's robustness.

In this work, we focus on constructing certifiers to find *certified regions* of the input neighborhood where the model is *guaranteed* to give the correct prediction, and on using such certifiers to train a model to be *provably* robust against adversarial attacks. In this context, complete certifiers can either guarantee the absence of an adversary or construct an adversarial example given an adversarial budget. They are typically built on either Satisfiability Modulo Theories (SMT) (Katz et al., 2017) or mixed integer programming (MIP) (Tjeng et al., 2017; Xiao et al., 2018). The major disadvantages of complete certifiers are their super-polynomial complexity and applicability to only piecewise linear activation functions, such as ReLU. By contrast, incomplete certifiers are faster, more widely applicable but more conservative in terms of certified regions because they rely on approximations. Techniques such as linear approximation (Kolter & Wong, 2017; Wong et al., 2018; Zhang et al., 2018; Weng et al., 2018), symbolic interval analysis (Wang et al., 2018b), abstract transformers (Gehr

et al., 2018; Singh et al., 2018; 2019), semidefinite programming (SDP) (Raghunathan et al., 2018a;b) and randomized smoothing (Cohen et al., 2019; Salman et al., 2019a) are exploited to offer better certified robustness. Some of these methods enable training provably robust models (Raghunathan et al., 2018a; Kolter & Wong, 2017; Wong et al., 2018; Cohen et al., 2019; Salman et al., 2019a) by optimizing the parameters so as to maximize the area of the certified regions.

While effective, all the above-mentioned methods, in their vanilla version, only provide binary results given a *fixed* adversarial budget. That is, if a data point is certified, it is guaranteed to be robust in the entire given adversarial budget; otherwise no guaranteed adversary-free region is estimated. To overcome this and search for the optimal adversarial budget that can be certified, Kolter & Wong (2017); Weng et al. (2018); Zhang et al. (2018) use either Newton's method or binary search. By contrast, Croce et al. (2018) take advantage of the geometric property of ReLU networks and gives finer-grained robustness guarantees. Given the piecewise linear ReLU function, any input is located in a polytope where the ReLU network can be considered linear. Based on geometry, robustness guarantees can thus be calculated using the input distance to the polytope boundary and the decision boundary constraints. Unfortunately, in practice, the resulting certified bounds are trivial because such polytopes are very small even for robust models. Nevertheless, Croce et al. (2018) introduce a regularization scheme based on these bounds to effectively train provably robust models.

In this paper, we construct a stronger certifier, as well as a regularization scheme to train provably robust models. To this end, building upon the geometry-inspired idea of Croce et al. (2018), we estimate a linear bound of the model's output given a predefined adversarial budget. Then, the condition to guarantee robustness inside this budget is also linear and forms a polyhedral envelope of the model's decision boundary. Similar to Croce et al. (2018), the region contained in the polyhedral envelope and the adversarial budget is certified to be adversary-free. We then further exploit this certifier to design a regularization framework to train provably robust neural network models.

In short, our contributions are twofold. First, we design a geometric-inspired certification method giving finer-grained one-shot bounds than its counterparts (Kolter & Wong, 2017; Weng et al., 2018; Zhang et al., 2018) with little overhead and applies to general activation functions. More importantly, we propose a regularization scheme to train provably robust models. Compared with existing provable training methods, which have been found to over-regularize the model (Zhang et al., 2019), our training framework yields competitive robustness guarantees while maintaining good performance on clean inputs. In the remainder of the paper, we call our certification method *Polyhedral Envelope Certifier (PEC)* and regularization scheme *Polyhedral Envelope Regularizer (PER)*.

## 2 PRELIMINARIES

### 2.1 NOTATION AND TERMINOLOGY

For simplicity, we discuss our approach using a standard $N$-layer fully-connected network. Note, however, that it straightforwardly extends to any network topology that can be represented by a directed acyclic graph (DAG) in the same way as in (Liu et al., 2019). A fully-connected network parameterized by $\{\mathbf{W}^{(i)}, \mathbf{b}^{(i)}\}_{i=1}^{N-1}$ can be expressed as

$$
\begin{aligned}
\mathbf{z}^{(i+1)} &= \mathbf{W}^{(i)}\hat{\mathbf{z}}^{(i)} + \mathbf{b}^{(i)} & i &= 1, 2, ..., N-1 \\
\hat{\mathbf{z}}^{(i)} &= \sigma(\mathbf{z}^{(i)}) & i &= 2, 3, ..., N-1
\end{aligned}
\tag{1}
$$

where $\mathbf{z}^{(i)}$ and $\hat{\mathbf{z}}^{(i)}$ are the pre- & post-activations of the $i$-th layer, respectively, and $\hat{\mathbf{z}}^{(1)} \stackrel{\text{def}}{=} \mathbf{x}$ is the input of the network. An $l_p$ norm-based adversarial budget $\mathcal{S}_\epsilon^{(p)}(\mathbf{x})$ is defined as the set $\{\mathbf{x}' | \|\mathbf{x}' - \mathbf{x}\|_p \le \epsilon\}$. $\mathbf{x}'$, $\mathbf{z}'^{(i)}$ and $\hat{\mathbf{z}}'^{(i)}$ represent the adversarial input and the corresponding pre- & post-activations. For layer $i$ having $n_i$ neurons, we have $\mathbf{W}^{(i)} \in \mathbb{R}^{n_{i+1} \times n_i}$ and $\mathbf{b}^{(i)} \in \mathbb{R}^{n_{i+1}}$.

Throughout this paper, underlines and bars are used to present lower and upper bounds of adversarial activations, respectively, i.e., $\underline{\mathbf{z}}^{(i)} \le \mathbf{z}'^{(i)} \le \bar{\mathbf{z}}^{(i)}$. A "+" or "−" subscript indicates the positive or negative part of a tensor. We use $[K]$ as the abbreviation for the set $\{1, 2, ..., K\}$.

### 2.2 MODEL LINEARIZATION

Given an adversarial budget $\mathcal{S}_\epsilon^{(p)}(\mathbf{x})$, we study the linear bound of the output logits $\mathbf{z}'^{(N)}$, given by

$$\mathbf{U}^{(N)}\mathbf{x}' + \mathbf{p}^{(N)} \leq \mathbf{z}'^{(N)} \leq \mathbf{V}^{(N)}\mathbf{x}' + \mathbf{q}^{(N)} \tag{2}$$

The linear coefficients introduced above can be calculated by iteratively estimating the bounds of intermediate layers and linearizing the activation functions. In Appendix A.1, we discuss this for several activation functions, including ReLU, sigmoid, tanh and arctan. Note that our method differs from Zhang et al. (2018) because we need the analytical form of the linear coefficients for training, which removes numerical methods such as binary search. The bounding algorithm trades off computational complexity and bound tightness. In this work, we study two such algorithms. One is based on Fast-Lin / CROWN (Weng et al., 2018; Zhang et al., 2018). It has tighter bounds but high computational complexity. Another is inspired by the interval bound propagation (IBP) (Gowal et al., 2018), which is faster but leads to looser bounds. The details of both algorithms are provided in Appendices A.2 and A.3. We briefly discuss the complexity of both algorithms in Section 5.

## 3 ALGORITHMS

### 3.1 ROBUSTNESS GUARANTEES BY POLYHEDRAL ENVELOPE

For an input point $\mathbf{x}$ with label $c \in [K]$, a sufficient condition to guarantee robustness is that the lower bounds of $\mathbf{z}'^{(N)}_c - \mathbf{z}'^{(N)}_i$ are positive for all $i \in [K]$. Here, we use a trick called *elision of the last layer* (Gowal et al., 2018) to merge the subtraction of $\mathbf{z}'^{(N)}_c$ and $\mathbf{z}'^{(N)}_i$ with the last linear layer and obtain the lower bound of $\mathbf{z}'^{(N)}_c - \mathbf{z}'^{(N)}_i$: $\underline{\mathbf{z}'^{(N)}_c - \mathbf{z}'^{(N)}_i} \overset{\text{def}}{=\joinrel=} \mathbf{U}_i\mathbf{x}' + \mathbf{p}_i$. Then, the sufficient condition to ensure robustness within a budget $\mathcal{S}^{(p)}_\epsilon(\mathbf{x})$ can be written as

$$\underline{\mathbf{z}'^{(N)}_c - \mathbf{z}'^{(N)}_i} = \mathbf{U}_i\mathbf{x}' + \mathbf{p}_i \geq 0 \quad \forall i \in [K] . \tag{3}$$

The constraint is trivial when $i = c$, so there are $K-1$ such linear constraints, corresponding to $K-1$ hyperplanes in the input space. Within the adversarial budget, these hyperplanes provide a polyhedral envelope of the true decision boundary. In the remainder of the paper, we use the term $d_{ic}$ to represent the distance between the input and the hyperplane defined in (3) and define $d_c = \min_{i \in [K], i \neq c} d_{ic}$ as the distance between the input and the polyhedral envelope boundary. The distance can be based on different $l_p$ norms, and $d_{ic} = 0$ when the input itself does not satisfy the inequality (3). Since (3) is a sufficient condition for robustness given the adversarial budget $\mathcal{S}^{(p)}_\epsilon(\mathbf{x})$, there is no adversarial examples in the intersection of $\mathcal{S}^{(p)}_\epsilon(\mathbf{x})$ and the polytope defined in (3).

The theorem below formalizes our robustness certification. We defer its proof to Appendix C.1.

**Theorem 1** (PEC in Unconstrained Cases). *Given a model $f : \mathbb{R}^{n_1} \to [K]$ and an input point $\mathbf{x}$ with label $c$, let $\mathbf{U}$ and $\mathbf{p}$ in (3) be calculated using a predefined adversarial budget $\mathcal{S}^{(p)}_\epsilon(\mathbf{x})$. Then, there is no adversarial example inside an $l_p$ norm ball of radius $d$ centered around $\mathbf{x}$, with $d = \min\{\epsilon, d_c\}$ where $d_{ic} = \max\left\{0, \frac{\mathbf{U}_i\mathbf{x} + \mathbf{p}_i}{\|\mathbf{U}_i\|_q}\right\}$. $l_q$ is the dual norm of the $l_p$ norm, i.e., $\frac{1}{p} + \frac{1}{q} = 1$.*

Based on Theorem 1, when $\epsilon < d_c$, PEC has the same robustness guarantees as KW (Kolter & Wong, 2017), Fast-Lin (Weng et al., 2018) and CROWN (Zhang et al., 2018) if we use the same model linearization method. When $0 < d_c < \epsilon$, KW / Fast-Lin / CROWN cannot certify the data point at all, while PEC still gives non-trivial robustness guarantees thanks to the geometric interpretability of the polyhedral envelope. Figure 1(a) compares the certified bounds of KW / Fast-Lin and PEC on a randomly picked input for different values of $\epsilon$.

Figure 1(b) shows a 2D sketch of the two cases mentioned above. If $\epsilon$ is too small, as in the left portion of the figure, the linear bounds in (3) are tight but only valid in a small region $\mathcal{S}^{(p)}_\epsilon(\mathbf{x})$. Therefore, the certified robustness is $\epsilon$ at most. If $\epsilon$ is too large, the linear bounds are valid in a larger region but more pessimistic because of linear approximation. [1] This is depicted by the right portion of the figure, where the distances between the input and the hyperplanes are smaller and the certified robustness is then $d_c$. The hyperplane segments inside the adversarial budget (green lines) never exceed the decision boundary (dark blue lines), by definition of the polyhedral envelope.

---

[1] The value of $d_{ic}$ monotonically decreases with the increase of $\epsilon$.

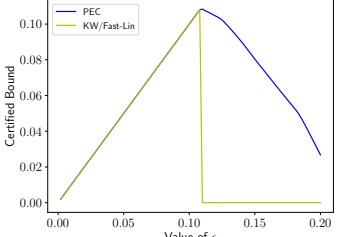
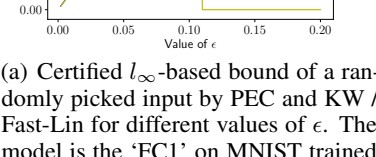
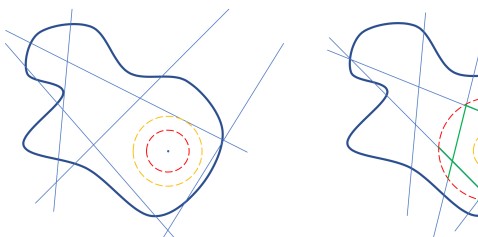

(a) Certified $l_\infty$-based bound of a randomly picked input by PEC and KW / Fast-Lin for different values of $\epsilon$. The model is the 'FC1' on MNIST trained by 'MMR+at' in Croce et al. (2018)

(b) 2D sketch of decision boundary (dark blue), hyperplane defined by (3) (light blue), adversarial budget (red), polyhedral envelope (green) in PEC. The distance between the input data and the hyperplanes is depicted by a yellow circle. The left and right portion correspond to $d_c$ bigger and smaller than $\epsilon$ respectively.

**Figure 1:** The two phases in PEC algorithm.

To search for the optimal $\epsilon$, i.e., the peak in Figure 1(a), Kolter & Wong (2017) use Newton's method, which is an expensive second-order method. By contrast, Weng et al. (2018); Zhang et al. (2018) use binary search, and our PEC can accelerate their strategy. The acceleration comes from the partial credit when $0 < d_c < \epsilon$, in which case we can rule out all values of $\epsilon$ below $d_c$ and obtain a better lower bound of the optimal $\epsilon$. A more detailed discussion is provided in Appendix B.1.

Under many circumstances, the input is constrained in a hypercube $[r^{(min)}, r^{(max)}]^{n_1}$. For example, for images with normalized pixel values, an attacker should not perturb the image out of the hypercube $[0, 1]^{n_1}$. Because of this constraint for the attacker, the certified regions become larger.

To obtain robustness guarantees in this scenario, we need to recalculate $d_{ic}$, which is now the distance between the input and the hyperplanes in (3) within the hypercube. The value of $d_{ic}$ then is the minimum of the optimization problem

$$\min_\Delta \|\Delta\|_p$$
$$s.t. \ \mathbf{a}\Delta + b \le 0, \ \Delta^{(min)} \le \Delta \le \Delta^{(max)} \tag{4}$$

where, to simplify the notation, we define $\mathbf{a} = \mathbf{U}_i$, $b = \mathbf{U}_i\mathbf{x} + \mathbf{p}$, $\Delta^{(min)} = r^{(min)} - \mathbf{x}$ and $\Delta^{(max)} = r^{(max)} - \mathbf{x}$. When $b \le 0$, the minimum is obviously 0 as the optimal $\Delta$ is an all-zero vector. In this case, either we cannot certify the input at all or even the clean input is misclassified. When $b > 0$, by Hölder's inequality, $\mathbf{a}\Delta + b \ge -\|\Delta\|_p\|\mathbf{a}\|_q + b$, with equality reached when $\Delta^p$ and $\mathbf{a}^q$ are collinear. Based on this, the optimal $\Delta$ of minimum $l_p$ norm to satisfy $\mathbf{a}\Delta + b \le 0$ is

$$\widehat{\Delta}_i = -\frac{b}{\|\mathbf{a}\|_q^q}\text{sign}(\mathbf{a}_i)\,|\mathbf{a}_i|^{\frac{q}{p}} \ , \tag{5}$$

where $\text{sign}(\cdot)$ returns $+1$ for positive numbers and $-1$ for negative numbers.

To satisfy the constraint $\Delta^{(min)} \le \Delta \le \Delta^{(max)}$, we use a greedy algorithm that approaches this goal progressively. That is, we first calculate the optimal $\widehat{\Delta}$ based on Equation (5) and check if the constraint $\Delta^{(min)} \le \Delta \le \Delta^{(max)}$ is satisfied. For the elements where it is not, we clip their values within $[\Delta^{(min)}, \Delta^{(max)}]$ and keep them fixed. We then optimize the remaining elements of $\Delta$ in the next iteration and repeat this process until the constraint is satisfied for all elements. The pseudo-code is provided as Algorithm 2 in Appendix B.2. The output of Algorithm 2 is the optimum of Problem (4), i.e., $d_{ic}$. We provide the proof in Appendix C.2. The certified bound in this constrained case is then $\min\{\epsilon, d_c\}$ just as demonstrated by Theorem 1.

### 3.2 GEOMETRY-INSPIRED REGULARIZATION

As in (Croce et al., 2018), we can incorporate our certification bounds in the training process so as to obtain more robust models. To this end, we design a regularization term that encourages larger values of $d_c$. We first introduce the *signed distance* $\tilde{d}_{ic}$: when $d_{ic} > 0$, the clean input satisfies (3) and

$\tilde{d}_{ic} = d_{ic}$; when $d_{ic} = 0$, the clean input does not satisfy (3) and there is no certified region; $\tilde{d}_{ic}$ in this case is a negative number whose absolute value is the distance between the input and the hyperplane defined in (3). If the input is unconstrained, we have $\tilde{d}_{ic} = \frac{\mathbf{U}_i \mathbf{x} + \mathbf{p}_i}{\|\mathbf{U}_i\|_q}$. Otherwise, following the notation of (4), $\tilde{d}_{ic} = sign(b)\|\widehat{\Delta}\|_p$, where $\widehat{\Delta} = \arg\min_\Delta \|\Delta\|_p$, $s.t.$ $\mathbf{a}\Delta + b = 0, \Delta^{(min)} \leq \Delta \leq \Delta^{(max)}$. This problem can be solved by a greedy algorithm similar to the one in Section 3.1.

Now, we sort $\{\tilde{d}_{ic}\}_{i=0, i\neq c}^{K-1}$ as $\tilde{d}_{j_0 c} \leq \tilde{d}_{j_1 c} \leq ... \leq \tilde{d}_{j_{K-3} c} \leq \tilde{d}_{j_{K-2} c}$ and then define the *Polyhedral Envelope Regularization (PER)* term, based on smallest $T$ distances, as

$$\text{PER}(\mathbf{x}, \alpha, \gamma, T) = \gamma \sum_{i=0}^{T-1} \max\left(0, 1 - \frac{\tilde{d}_{j_i c}}{\alpha}\right). \tag{6}$$

Note that, following Croce et al. (2018), to accelerate training, we take into account the smallest $T$ distances. When $\tilde{d}_{j_i c} \geq \alpha$, the distance is considered big enough, so the corresponding term will not contribute to the gradient of the model parameters. This avoids over-regularization and allows us to maintain accuracy on clean inputs. In practice, we do not activate PER in the early training stages, when the model is not well trained and the corresponding polyhedral envelope is meaningless. Such 'warm up' trick is commonly used in the deep learning practice (Gotmare et al., 2018).

Note that we can further incorporate PER with adversarial training in a similar way to Croce et al. (2018). The only change w.r.t. the original PER consists of replacing $\tilde{d}_{j_i c}$ with $\tilde{d}'_{j_i c}$, the distance between the polyhedral envelope and the adversarial example generated by PGD (Madry et al., 2017) instead of the clean input. We call the corresponding method *PER+at*.

Calculating the polyhedral envelope is expensive in terms of both computation and memory because of the need to obtain linear bounds of the output logits. To prevent a prohibitive computational and memory overhead, we use the stochastic robust approximation of Wang et al. (2018a). For a mini-batch of size $B$, we only calculate the PER or PER+at regularization term for $B' < B$ instances sub-sampled from this mini-batch. Moosavi-Dezfooli et al. (2017) empirically points out the geometric correlation of high-dimensional decision boundaries near the data manifold. Thus, in practice, a $B'$ much smaller than $B$ provides a good approximation of the full-batch regularization.

## 4 EXPERIMENTS

To validate the theory and algorithms above, we conducted several experiments on two popular image classification benchmarks: MNIST and CIFAR10. Each of these experiments can be completed on a single GPU machine within hours. We will make the code and checkpoints publicly available.

### 4.1 TRAINING AND CERTIFYING RELU NETWORKS

As the main experiment, we first demonstrate the benefits of our approach over the state-of-the-art training and certification methods. To this end, we use the same model architectures as in Croce et al. (2018); Kolter & Wong (2017): **FC1**, which is a fully-connected network with one hidden layer of 1024 neurons; and **CNN**, which has two convolutional layers followed by two fully-connected layers. For this set of experiments, all activation functions are ReLU.

We focus on training and certification method based on model linearization and compare 7 different training schemes using 5 evaluation metrics under the same and fixed adversarial budgets. All the certification results here are *one-shot* results as the binary search for optimal $\epsilon$ is not used. The 7 training schemes are plain training (plain), adversarial training (at) (Madry et al., 2017), KW (Kolter & Wong, 2017), MMR (Croce et al., 2018), MMR plus adversarial training (MMR+at) (Croce et al., 2018), and our PER and PER+at of Section 3.2. For certification, we compare Fast-Lin (Weng et al., 2018), KW (Kolter & Wong, 2017) and our PEC introduced in Section 3.1. KW is algorithmically equivalent to Fast-Lin with *elision of the last layer* (Salman et al., 2019b), so we consider them as one certification method. Based on the discussions in Section 3.1, all three methods have the same certified robust error, which is the proportion of the inputs whose certified regions are smaller than the adversarial budget. Therefore, we report the clean test error (CTE), empirical robust error against PGD (PGD), the certified robust error (CRE Lin), the average certified bound by Fast-Lin / KW (ACB

| | CTE (%) | PGD (%) | CRE Lin (%) | ACB KW | ACB PEC | CTE (%) | PGD (%) | CRE Lin (%) | ACB KW | ACB PEC |
|---|---|---|---|---|---|---|---|---|---|---|
| | **MNIST - FC1, ReLU, $l_\infty, \epsilon = 0.1$** | | | | | **MNIST - FC1, ReLU, $l_2, \epsilon = 0.3$** | | | | |
| plain | 1.99 | 98.37 | 100.00 | 0.0000 | 0.0000 | 1.99 | 9.81 | 40.97 | 0.1771 | 0.2300 |
| at | 1.42 | 9.00 | 97.94 | 0.0021 | 0.0099 | 1.35 | 2.99 | 14.85 | 0.2555 | 0.2684 |
| KW | 2.26 | 8.59 | 12.91 | 0.0871 | 0.0928 | 1.23 | 2.70 | 4.91 | 0.2853 | 0.2892 |
| MMR | 2.11 | 17.82 | 33.75 | 0.0663 | 0.0832 | 2.40 | 5.88 | 7.76 | 0.2767 | 0.2845 |
| MMR+at | 2.04 | 10.39 | 17.64 | 0.0824 | 0.0905 | 1.77 | 3.76 | 5.68 | 0.2830 | 0.2880 |
| PER | 1.60 | 7.45 | 11.71 | 0.0883 | 0.0935 | 1.26 | 2.44 | 5.35 | 0.2840 | 0.2888 |
| PER+at | 1.81 | 7.73 | 12.90 | 0.0871 | 0.0925 | 0.67 | 1.40 | 4.84 | 0.2855 | 0.2910 |
| | **MNIST - CNN, ReLU, $l_\infty, \epsilon = 0.1$** | | | | | **MNIST - CNN, ReLU, $l_2, \epsilon = 0.3$** | | | | |
| plain | 1.28 | 85.75 | 100.00 | 0.0000 | 0.0000 | 1.28 | 4.93 | 100.00 | 0.0000 | 0.0000 |
| at | 1.02 | 4.75 | 91.91 | 0.0081 | 0.0189 | 1.12 | 2.50 | 100.00 | 0.0000 | 0.0000 |
| KW | 1.21 | 3.03 | 4.44 | 0.0956 | 0.0971 | 1.11 | 2.05 | 5.84 | 0.2825 | 0.2861 |
| MMR | 1.65 | 6.09 | 11.56 | 0.0884 | 0.0928 | 2.57 | 5.49 | 10.03 | 0.2699 | 0.2788 |
| MMR+at | 1.19 | 3.35 | 9.49 | 0.0905 | 0.0939 | 1.73 | 3.22 | 9.46 | 0.2716 | 0.2780 |
| PER | 1.44 | 3.44 | 5.13 | 0.0949 | 0.0965 | 1.02 | 1.87 | 5.04 | 0.2849 | 0.2882 |
| PER+at | 0.50 | 2.02 | 4.85 | 0.0952 | 0.0969 | 0.43 | 0.91 | 5.43 | 0.2837 | 0.2878 |
| | **CIFAR10 - CNN, ReLU, $l_\infty, \epsilon = 2/255$** | | | | | **CIFAR10 - CNN, ReLU, $l_2, \epsilon = 0.1$** | | | | |
| plain | 24.62 | 86.29 | 100.00 | 0.0000 | 0.0000 | 23.29 | 47.39 | 100.00 | 0.0000 | 0.0000 |
| at | 27.04 | 48.53 | 85.36 | 0.0011 | 0.0015 | 25.84 | 35.81 | 99.96 | 0.0000 | 0.0000 |
| KW | 39.27 | 48.16 | 53.81 | 0.0036 | 0.0040 | 40.24 | 43.87 | 48.98 | 0.0510 | 0.0533 |
| MMR | 34.59 | 57.17 | 69.28 | 0.0024 | 0.0032 | 40.93 | 50.57 | 57.07 | 0.0429 | 0.0480 |
| MMR+at | 35.36 | 49.27 | 59.91 | 0.0031 | 0.0037 | 37.78 | 43.98 | 53.33 | 0.0467 | 0.0502 |
| PER | 39.21 | 50.98 | 57.45 | 0.0033 | 0.0038 | 34.10 | 52.54 | 63.42 | 0.0369 | 0.0465 |
| PER+at | 28.87 | 43.55 | 56.59 | 0.0034 | 0.0040 | 25.76 | 33.47 | 46.74 | 0.0533 | 0.0580 |

**Table 1:** Results for 7 training schemes and 5 evaluation schemes for ReLU networks. The best and second-best results among robust training methods (*plain* and *at* excluded) are highlighted by dark gray and light gray.

KW) and by PEC (ACB PEC) on the test set. We do not report the certified bound by MMR (Croce et al., 2018), because, in practice, it only gives trivial results. As a matter of fact, Croce et al. (2018) emphasize their training method and report certification results using KW and MIP (Tjeng et al., 2017). For methods that are not based on model linearization, such as IBP (Gowal et al., 2018) and CROWN-IBP (Zhang et al., 2019), we put more results and discussions in Table 5 of Appendix D.2.1.

We downloaded the KW, MMR and MMR+at models from the checkpoints provided by Croce et al. (2018).[2] To be fair, we use the same model linearization method for our method. For CNN models, we use the *warm up* trick consisting of performing adversarial training before adding our PER or PER+at regularization term. The running time overhead of pre-training is negligible compared with computing the regularization term. We train all models for 100 epochs and provide the detailed hyper-parameter settings in Appendix D.1.

We consider robustness in both the $l_\infty$ and $l_2$ case, and the attackers are constrained to perturb the images within $[0, 1]^{n_1}$. Table 1 shows the results with the same adversarial budget used in Croce et al. (2018). Consistently with Section 3.1, our geometry-inspired PEC has better average certified bounds than Fast-Lin / KW given the same adversarial budget. For CIFAR10 model against $l_\infty$ attack, there are $10\% - 20\%$ test points not certified by Fast-Lin / KW but have non-trivial bounds by PEC. Figure 5 of Appendix D.2.1 shows the distribution of these certified bounds. Among the training methods, PER / PER+at give better performance in all cases than MMR / MMR+at. When compared with KW, their certified robust error is comparable, but PER+at has better clean test accuracy, particularly on CIFAR10. In fact, PER+at maintains a similar clean test error to adversarial training (at) while achieving competitive certified robust error. In other words, a model trained by PER+at is not as over-regularized as other training methods for provable robustness. Figure 6 of Appendix D.2.1 shows the parameter value distribution of models trained by KW/MMR+at/PER+at. It is clear that parameters of PER+at models have the largest norms and thus better preserve the model capacity. The better performance of PER+at over PER, and of MMR+at over MMR, evidences the benefits of augmenting the training data with adversarial examples.

---

[2]Repository: https://github.com/max-andr/provable-robustness-max-linear-regions.

To demonstrate the trade-off between clean test error and certified robust error, we evaluate our approach with different regularizer strength $\gamma$ in (6). Figure 2 shows the results of PER+at in the $l_2$ case for CIFAR10. When $\gamma$ is small, PER+at becomes similar to adversarial training (at) and has low clean test error but high certified robust error. As $\gamma$ grows, the model is increasingly regularized towards large polyhedral envelopes, which inevitably hurts the performance on the clean input. By contrast, the certified robust error first decreases and then increases. This is because training is numerically more difficult when $\gamma$ is too large and the model is over-regularized. The results of KW are shown as horizontal dashed lines for comparison, which evidences that PER+at is less over-regularized in general than KW, with much lower clean test error for the same certified robust error.

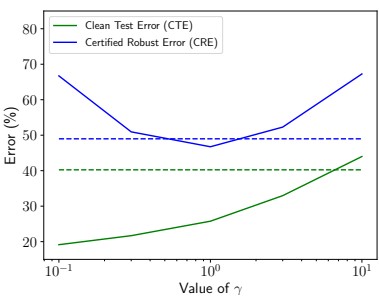

**Figure 2:** CTE and CRE for different values of $\gamma$ in PER+at. The results of KW, for reference, are the horizontal dashed lines.

The experimental results of PER / PER+at in Table 1 rely on the same model linearization methods as Fast-Lin / KW. Since our framework is applicable to any linearization strategy, we present the results using IBP-inspired linearization in Table 6 of Appendix D.2.2. Compared with Table 1, we find IBP-inspired linearization has competitive performance during training, although it leads to a looser polyhedral envelope. In addition, IBP-inspired linearization has less memory consumption and computational complexity, so it is suitable for machines of limited computational resources.

## 4.2 Training and Certifying Non-ReLU Networks

To validate our method's applicability to non-ReLU networks, we replace the ReLU function in FC1 with either sigmoid, or tanh, or arctan function. While Wong et al. (2018) claim that their methods apply to non-ReLU networks, their main contribution is rather the extension of KW to a broader set of network architectures, and their public code [3] does not support non-ReLU activations. Instead, we thus compare the average certified bound of CROWN (Zhang et al., 2018) (ACB CRO), which generalizes Fast-Lin to non-ReLU functions, with our PEC (ACB PEC). Since any linearization of activation functions can be plugged in both methods [4], we use the method in Appendix A.1.

|  | CTE (%) | PGD (%) | CRE Lin (%) | ACB CRO | ACB PEC | CTE (%) | PGD (%) | CRE Lin (%) | ACB CRO | ACB PEC |
|---|---|---|---|---|---|---|---|---|---|---|
| | **MNIST - FC1, Sigmoid, $l_\infty, \epsilon = 0.1$** | | | | | **MNIST - FC1, Sigmoid, $l_2, \epsilon = 0.3$** | | | | |
| plain | 2.04 | 97.80 | 100.00 | 0.0000 | 0.0000 | 2.01 | 10.25 | 30.78 | 0.2077 | 0.2539 |
| at | 1.78 | 10.05 | 98.52 | 0.0015 | 0.0055 | 1.65 | 3.48 | 7.50 | 0.2775 | 0.2839 |
| PER | 2.29 | 8.44 | 15.75 | 0.0843 | 0.0908 | 1.19 | 2.36 | 7.08 | 0.2787 | 0.2849 |
| PER+at | 2.46 | 8.14 | 15.00 | 0.0850 | 0.0906 | 0.67 | 1.25 | 6.93 | 0.2792 | 0.2868 |
| | **MNIST - FC1, Tanh, $l_\infty, \epsilon = 0.1$** | | | | | **MNIST - FC1, Tanh, $l_2, \epsilon = 0.3$** | | | | |
| plain | 2.00 | 99.57 | 100.00 | 0.0000 | 0.0000 | 1.94 | 16.46 | 61.66 | 0.1150 | 0.1789 |
| at | 1.28 | 8.89 | 99.98 | 0.0000 | 0.0001 | 1.36 | 3.02 | 12.35 | 0.2630 | 0.2735 |
| PER | 2.66 | 9.31 | 16.02 | 0.0840 | 0.0908 | 1.39 | 2.73 | 7.56 | 0.2773 | 0.2834 |
| PER+at | 2.73 | 8.45 | 14.99 | 0.0850 | 0.0907 | 0.81 | 1.76 | 11.80 | 0.2646 | 0.2745 |
| | **MNIST - FC1, ArcTan, $l_\infty, \epsilon = 0.1$** | | | | | **MNIST - FC1, ArcTan, $l_2, \epsilon = 0.3$** | | | | |
| plain | 1.95 | 99.83 | 100.00 | 0.0000 | 0.0000 | 2.09 | 23.00 | 62.82 | 0.1115 | 0.1869 |
| at | 1.33 | 8.88 | 99.92 | 0.0001 | 0.0005 | 1.56 | 3.04 | 10.10 | 0.2697 | 0.2784 |
| PER | 2.78 | 9.55 | 16.75 | 0.0833 | 0.0904 | 1.42 | 2.66 | 7.81 | 0.2766 | 0.2828 |
| PER+at | 2.98 | 8.96 | 15.64 | 0.0844 | 0.0901 | 0.83 | 1.61 | 11.38 | 0.2659 | 0.2758 |

**Table 2:** Results of FC1 models with non-ReLU activation functions on MNIST.

According to Table 2, in contrast to plain training and adversarial training, our PER / PER+at can train provably robust models, despite a slightly lower performance on the clean test data. For certification, PEC has better average certified bounds than CROWN given the same adversarial budget.

---

[3]Repository: https://github.com/locuslab/convex_adversarial

[4]For certification, there is no need to require an analytical form of the linearization.

### 4.3 FINDING THE OPTIMAL BUDGET

To obtain the biggest certified bound, we need to search for the optimal value of $\epsilon$, i.e., the peak in Figure 1(a). While Kolter & Wong (2017) use Newton's method to solve a constrained optimization problem, which is expensive, Fast-Lin / CROWN (Weng et al., 2018; Zhang et al., 2018) apply a binary search strategy to find the optimal $\epsilon$.

To validate the claim in Section 3.1 that PEC can find the optimal $\epsilon$ faster than Fast-Lin / CROWN, we compare the average number of iterations needed to find the optimal value given a required precision $\epsilon_\Delta$. Using $\underline{\epsilon}$ and $\bar{\epsilon}$ to define the initial lower and upper estimates of the optimal value, then we need $\lceil \log_2 \frac{\bar{\epsilon}-\underline{\epsilon}}{\epsilon_\Delta} \rceil$ steps of bound calculation to obtain the optimal value by binary search in Fast-Lin / CROWN. By contrast, the number of iterations needed by PEC depends on the model to certify. The pseudo-code of the search algorithm is provided in Appendix B.1.

We briefly discuss the experimental results here, and defer their details to Appendix D.2.3. Note that, because PEC has almost no computational overhead compared with Fast-Lin and CROWN, the number of iterations reflects the running time to obtain the optimal certified bounds. Altogether, our results show that PEC can save on average 25% of the running time for FC1 models with ReLU, 15% for FC1 models with non-ReLU activations and 10% for CNN models.

## 5 DISCUSSION

Let us consider an $N$-layer neural network model with $k$-dimensional output and $m$-dimensional input. For simplicity, let each hidden layer have $n$ neurons and $n \gg \max\{k, m\}$. In this context, the bounding algorithm based on Fast-Lin / CROWN need $\mathcal{O}(N^2 n^3)$ FLOPs to obtain linear bounds of output logits. However, the complexity can be reduced to $\mathcal{O}(Nn^2 m)$ at the cost of bound tightness when we use the IBP-inspired algorithm. In MMR (Croce et al., 2018), the complexity to calculate the expression of the input's linear region is also $\mathcal{O}(Nn^2 m)$. On the training side, KW needs a back-propagation to calculate the bound, which costs $\mathcal{O}(Nn^2)$. MMR needs to calculate the distance between the input and $\mathcal{O}(Nn)$ hyper-planes, costing $\mathcal{O}(Nnm)$ FLOPs, while PER only calculates $\mathcal{O}(k)$ hyperplanes, thus requiring $\mathcal{O}(km)$ FLOPs. Overall, we can see that the estimation of the output logits or decision boundary dominates the complexity of all training algorithms and is the main barrier towards scalable provably robust training. The complexity of PER is similar to Fast-Lin / CROWN if we use the same model linearization method as theirs. When we use an IBP-inspired algorithm instead, the complexity of PER is similar to MMR. Note that the FLOP complexity of PGD with $h$ iterations is $\mathcal{O}(Nn^2 h)$ and typically $h \ll \min\{m, n\}$, so the overhead of the warm up phase is negligible for PER training.

No matter which linearization method we use, the bounds of the output logits inevitably become looser for deeper networks, which can be a problem for large models. Furthermore, the linear approximation implicitly favors the $l_\infty$ norm over other $l_p$ norms because the intermediate bounds are calculated in an elementwise manner (Liu et al., 2019). As a result, our method performs better in $l_\infty$ cases than in $l_2$ cases. Designing a training algorithm with scalable and tight certified robustness is highly non-trivial and worth further exploration.

## 6 CONCLUSION

In this paper, we have studied the robustness of neural networks from a geometric perspective. In our framework, linear bounds are estimated for the model's output under an adversarial budget. Then, the polyhedral envelope resulting from the linear bounds allows us to obtain quantitative robustness guarantees. Our certification method can give non-trivial robustness guarantees to more data points than existing methods. Furthermore, we have shown that our certified bounds can be turned into a geometry-inspired regularization scheme that enables training provably robust models. Compared with existing methods, our proposed framework can be applied to neural networks with general activation functions. Unlike many over-regularized methods, it can achieve provable robustness at very little loss in clean accuracy. Extending this framework to larger networks will be the focus of our future research.

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

## A  MODEL LINEARIZATION

### A.1  LINEARIZATION OF ACTIVATION FUNCTIONS

In this section, we discuss the choice of $d$, $l$ and $h$ in the linear approximation $dx + l \leq \sigma(x) \leq dx + h$ for activation function $\sigma$ when $x \in [\underline{x}, \bar{x}]$. Method used here is slightly different from Zhang et al. (2018). First, the slope of the linear upper and lower bound is the same for simplicity. Second, all coefficients need to have analytical form because we need to calculate the gradient based on them during training. Note that Zhang et al. (2018) use binary search for optimal $d$, $l$, $h$.

#### A.1.1  RELU

As Figure 3 shows, the linear approximation for ReLU $\sigma(x) = \max(0, x)$, which is convex, is:

$$d = \begin{cases} 0 & \underline{x} \leq \bar{x} \leq 0 \\ \dfrac{\bar{x}}{\bar{x} - \underline{x}} & \underline{x} < 0 < \bar{x}, l = 0, h = \begin{cases} 0 & \underline{x} \leq \bar{x} \leq 0 \\ -\dfrac{\underline{x}\bar{x}}{\bar{x} - \underline{x}} & \underline{x} < 0 < \bar{x} \\ 1 & 0 \leq \underline{x} \leq \bar{x} \end{cases} \\ 0 & 0 \leq \underline{x} \leq \bar{x} \end{cases} \tag{7}$$

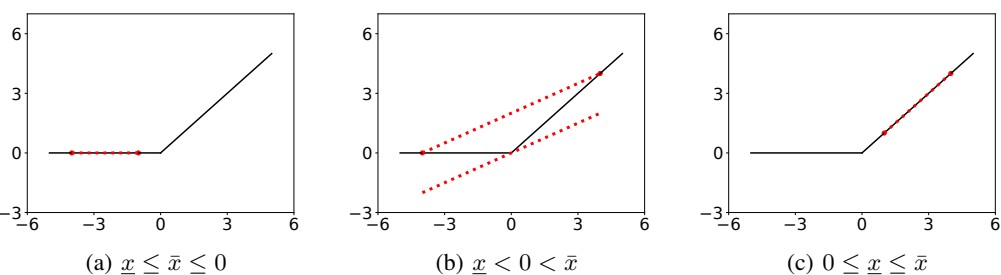

(a) $\underline{x} \leq \bar{x} \leq 0$     (b) $\underline{x} < 0 < \bar{x}$     (c) $0 \leq \underline{x} \leq \bar{x}$

**Figure 3:** The linearization of ReLU function in all scenarios.

#### A.1.2  SIGMOID, TANH, ARCTAN

Unlike ReLU function, Tanh function $\sigma(x) = \frac{e^{2x}-1}{e^{2x}+1}$, Sigmoid function $\sigma(x) = \frac{1}{1+e^{-x}}$ and Arctan function $\sigma(x) = arctan(x)$ are not convex. However, these three functions are convex when $x < 0$ and concave when $x > 0$. (Left and right sub-figures of Figure 4) Therefore, when $\underline{x} \leq \bar{x} \leq 0$ or $0 \leq \underline{x} \leq \bar{x}$, we can easily obtain the tight linear approximation. When $\underline{x} \leq 0 \leq \bar{x}$, we do not use the binary research to obtain the tight linear approximation like Zhang et al. (2018), because the results would not have an analytical form in this way. Instead, we first calculate the slope between the two ends i.e. $d = \frac{\sigma(\bar{x})-\sigma(\underline{x})}{\bar{x}-\underline{x}}$. Then, we bound the function by two tangent lines of the same slope as $d$ (middle sub-figure of Figure 4).

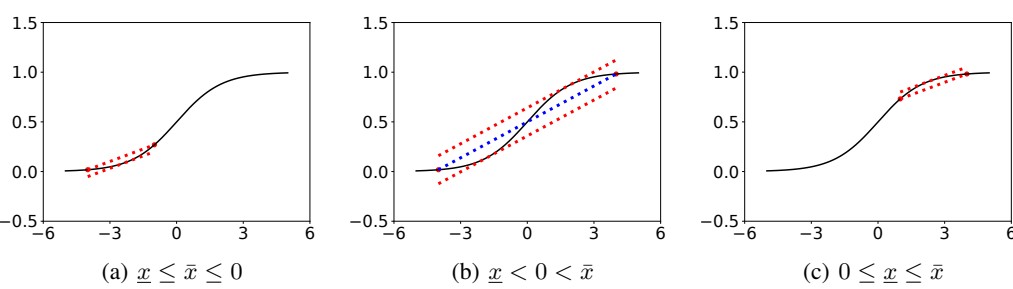

(a) $\underline{x} \leq \bar{x} \leq 0$     (b) $\underline{x} < 0 < \bar{x}$     (c) $0 \leq \underline{x} \leq \bar{x}$

**Figure 4:** The linearization of Sigmoid function in all scenarios.

For Tanh, Sigmoid and Arctan, we can calculate the coefficient of the linear approximation.

$$d = \frac{\sigma(\bar{x}) - \sigma(\underline{x})}{\bar{x} - \underline{x}}, l = \begin{cases} \sigma(t_1) - t_1 d & \underline{x} < 0 \\ \dfrac{\bar{x}\sigma(\underline{x}) - \underline{x}\sigma(\bar{x})}{\bar{x} - \underline{x}} & 0 \leq \underline{x} \leq \bar{x} \end{cases}, h = \begin{cases} \dfrac{\bar{x}\sigma(\underline{x}) - \underline{x}\sigma(\bar{x})}{\bar{x} - \underline{x}} & \underline{x} \leq \bar{x} \leq 0 \\ \sigma(t_2) - t_2 d & 0 < \bar{x} \end{cases} \quad (8)$$

The coefficient $t_1 < 0 < t_2$ are the position of tangent points on both sides of origin. The definitions of $t_1$ and $t_2$ for different activation functions provided in Table 3.

| $\sigma$ | Tanh | Sigmoid | Arctan |
|---|---|---|---|
| $t_1$ | $\frac{1}{2}\log\frac{-(d-2)-2\sqrt{1-d}}{d}$ | $-\log\frac{-(2d-1)+\sqrt{1-4d}}{2d}$ | $-\sqrt{\frac{1}{d}-1}$ |
| $t_2$ | $\frac{1}{2}\log\frac{-(d-2)+2\sqrt{1-d}}{d}$ | $-\log\frac{-(2d-1)-\sqrt{1-4d}}{2d}$ | $\sqrt{\frac{1}{d}-1}$ |

**Table 3:** Definition of $t_1$ and $t_2$ for different activation functions.

## A.2 BOUNDS BASED ON FAST-LIN / CROWN

Based on the linear approximation of activation functions above, we have $\mathbf{D}^{(i)}\mathbf{z}'^{(i)} + \mathbf{l}^{(i)} \leq \sigma(\mathbf{z}'^{(i)}) \leq \mathbf{D}^{(i)}\mathbf{z}'^{(i)} + \mathbf{u}^{(i)}$ where $\mathbf{D}^{(i)}$ is a diagonal matrix and $\mathbf{l}^{(i)}$, $\mathbf{u}^{(i)}$ are vectors. We can re-write this formulation as follows:

$$\exists \mathbf{D}^{(i)}, \mathbf{l}^{(i)}, \mathbf{u}^{(i)} : \forall \mathbf{z}'^{(i)} \in [\underline{\mathbf{z}}^{(i)}, \bar{\mathbf{z}}^{(i)}], \exists \mathbf{m}^{(i)} \in [\mathbf{l}^{(i)}, \mathbf{u}^{(i)}] \ s.t. \sigma(\mathbf{z}'^{(i)}) = \mathbf{D}^{(i)}\mathbf{z}'^{(i)} + \mathbf{m}^{(i)} \quad (9)$$

We plug (9) into (1) and the expression of $\mathbf{z}'^{(i)}$ can be rewritten as

$$\begin{aligned}
\mathbf{z}'^{(i)} &= \mathbf{W}^{(i-1)}(\sigma(\mathbf{W}^{(i-2)}(...\sigma(\mathbf{W}^{(1)}\hat{\mathbf{z}}'^{(1)} + \mathbf{b}^{(1)})...) + \mathbf{b}^{(i-2)})) + \mathbf{b}^{(i-1)} \\
&= \mathbf{W}^{(i-1)}(\mathbf{D}^{(i-1)}(\mathbf{W}^{(i-2)}(...\mathbf{D}^{(2)}(\mathbf{W}^{(1)}\mathbf{x}' + \mathbf{b}^{(1)}) + \mathbf{m}^{(2)}...) + \mathbf{b}^{(i-2)}) + \mathbf{m}^{(i-1)}) + \mathbf{b}^{(i-1)} \\
&= \left(\prod_{k=1}^{i-1} \mathbf{W}^{(k)}\mathbf{D}^{(k)}\right)\mathbf{W}^{(1)}\mathbf{x}' + \sum_{j=1}^{i-1}\left(\prod_{k=j+1}^{i-1}\mathbf{W}^{(k)}\mathbf{D}^{(k)}\right)\mathbf{b}^{(j)} + \sum_{j=2}^{i-1}\left(\prod_{k=j+1}^{i-1}\mathbf{W}^{(k)}\mathbf{D}^{(k)}\right)\mathbf{W}^{(j)}\mathbf{m}^{(j)}.
\end{aligned}$$
$$(10)$$

This is a linear function w.r.t $\mathbf{x}'$ and $\{\mathbf{m}^{(j)}\}_{j=2}^{i-1}$. Once given the perturbation budget $\mathcal{S}_\epsilon^{(p)}(\mathbf{x})$ and the bounds of $\{\mathbf{m}^{(j)}\}_{j=2}^{i-1}$, we can calculate the bounds of $\mathbf{z}'^{(i)}$ and the bias term in (10). This process can be repeated until we obtain the bound of output logits (2). The derivation here is the same as Weng et al. (2018); Liu et al. (2019), we encourage interested readers to check these works for details.

## A.3 BOUNDS INSPIRED BY INTERVAL BOUND PROPAGATION

Interval Bound Propagation (IBP) introduced in Gowal et al. (2018) is a simple and scalable method to estimate the bounds of each layer in neural networks. IBP is much faster than algorithm introduced in Appendix A.2 because the bounds of any intermediate layer are calculated only based on the information of its immediate previous layer. Therefore, the bounds are propagated just like the inference of network models, which cost only $\mathcal{O}(N)$ matrix-vector multiplications for a $N$-layer network defined in (1).

In our work, we need linear bounds of output logits in addition to general numeric bounds, so the linearization of activation functions defined in (9) is necessary. We define linear bounds $\mathbf{U}^{(i)}\mathbf{x}' + \mathbf{p}^{(i)} \leq \mathbf{z}'^{(i)} \leq \mathbf{U}^{(i)}\mathbf{x}' + \mathbf{q}^{(i)}$, $\widehat{\mathbf{U}}^{(i)}\mathbf{x}' + \hat{\mathbf{p}}^{(i)} \leq \hat{\mathbf{z}}'^{(i)} \leq \widehat{\mathbf{U}}^{(i)}\mathbf{x}' + \hat{\mathbf{q}}^{(i)}$. According to Section A.1, we use the same slope to linearize activation functions, so the slopes of both bounds here are the same. Plug (9) into this formulation, we have:

$$\widehat{\mathbf{U}}^{(i)} = \mathbf{D}^{(i)}\mathbf{U}^{(i)}, \ \hat{\mathbf{p}}^{(i)} = \mathbf{D}^{(i)}\mathbf{p}^{(i)} + \mathbf{l}^{(i)}, \ \hat{\mathbf{q}}^{(i)} = \mathbf{D}^{(i)}\mathbf{q}^{(i)} + \mathbf{u}^{(i)} \quad (11)$$

Here, we assume activation functions are monotonic increasing, so elements in $\mathbf{D}^{(i)}$ are non-negative. Similarity, compare the linear bounds of $\hat{\mathbf{z}}'^{(i)}$ and $\mathbf{z}'^{(i+1)}$, we have:

$$\mathbf{U}^{(i+1)} = \mathbf{W}^{(i)}\widehat{\mathbf{U}}^{(i)}, \ \mathbf{p}^{(i+1)} = \mathbf{W}_+^{(i)}\hat{\mathbf{p}}^{(i)} + \mathbf{W}_-^{(i)}\hat{\mathbf{q}}^{(i)} + \mathbf{b}^{(i)}, \ \mathbf{q}^{(i+1)} = \mathbf{W}_+^{(i)}\hat{\mathbf{q}}^{(i)} + \mathbf{W}_-^{(i)}\hat{\mathbf{p}}^{(i)} + \mathbf{b}^{(i)} \tag{12}$$

By definition, we have $\widehat{\mathbf{U}}^{(1)} = \mathbf{I}$ and $\hat{\mathbf{p}}^{(1)} = \hat{\mathbf{q}}^{(1)} = \mathbf{0}$. Apply (11) and (12) iteratively, we can obtain the values of coefficients $\mathbf{U}^{(N)}$, $\mathbf{V}^{(N)}$, $\mathbf{p}^{(N)}$ and $\mathbf{q}^{(N)}$ in (2).

## B  ALGORITHMS

### B.1  ALGORITHMS FOR SEARCH OPTIMAL $\epsilon$

The pseudo code for finding optimal $\epsilon$ is demonstrated as Algorithm 1 below. $\mathcal{M}, \mathbf{x}, \epsilon_\Delta, \underline{\epsilon}, \bar{\epsilon}$ represent the classification model, the input point, precision requirement, predefined estimate of lower bound and upper bound respectively. Typically, $\underline{\epsilon}$ is set $0$ and $\bar{\epsilon}$ is set a large value to which degree the image perturbation is perceptible. $f$ is a function mapping a model, an input point and a value of $\epsilon$ to a certified bound. $f$ is a generalization form of algorithm Fast-Lin, CROWN, PEC etc.

During the search for optimal $\epsilon$, the lower bound is updated by current certified bound, while the upper bound is updated only when current certified bound is smaller than the choice of $\epsilon$. In Fast-Lin and CROWN, we update either lower or upper bound in one iteration since the certified bound is either $0$ or current choice of $\epsilon$. However, it is possible for PEC to update both lower and upper bound in one iteration, which leads to a faster convergence of $\epsilon$.

---

**Algorithm 1:** Search for optimal value of $\epsilon$

**Input:** $\mathbf{x}, \underline{\epsilon}, \bar{\epsilon}, \epsilon_\Delta, f, \mathcal{M}$
Set the bounds of $\epsilon$: $\epsilon_{up} = \bar{\epsilon}, \epsilon_{low} = \underline{\epsilon}$;
**while** $\epsilon_{up} - \epsilon_{low} > \epsilon_\Delta$ **do**
  $\quad \epsilon_{try} = \frac{1}{2}(\epsilon_{low} + \epsilon_{up})$;
  $\quad \epsilon_{cert} = f(\mathcal{M}, \mathbf{x}, \epsilon_{try})$;
  $\quad$ Update lower bound: $\epsilon_{low} = \max\{\epsilon_{low}, \epsilon_{cert}\}$;
  $\quad$ **if** $\epsilon_{try} > \epsilon_{cert}$ **then**
    $\quad\quad |$ Update upper bound: $\bar{\epsilon} = \epsilon_{try}$;
  $\quad$ **end**
**end**
**Output:** $\frac{1}{2}(\epsilon_{low} + \epsilon_{up})$

---

### B.2  GREEDY ALGORITHM IN CONSTRAINED CASES

We provide the pseudo code of greedy algorithm to solve problem (4) as Algorithm 2 below.

To prevent prohibitive computational overhead, we set the maximum iteration number $I^{(max)}$ of the while-loop. If this maximum number is met and the iteration is broken, then the output is a lower bound of the solution in problem (4) and thus still a valid but suboptimal robustness guarantee. In practice, we set the maximum iteration number to be 20, although the while-loop breaks within 5 iterations in almost all cases.

## C  PROOFS

### C.1  PROOF OF THEOREM 1

*Proof.* We let $\mathbf{x}' = \mathbf{x} + \Delta$ be a point to break the condition (3).

---

**Algorithm 2:** Greedy algorithm to solve Problem (4).

---

**Input:** $\mathbf{x}, \mathbf{a}, b, \Delta^{(min)}, \Delta^{(max)}$ in (4) and maximum iteration $I^{(max)}$
Set of fixed elements $\mathcal{S}^{(f)} = \emptyset$;
Iteration number $i = 0$;
Calculate $\widehat{\Delta}$ according to (5);
**while** $\Delta^{(min)} \leq \widehat{\Delta} \leq \Delta^{(max)}$ *not satisfied and* $i < I^{(max)}$ **do**

$\quad$ Set of violated elements $\mathcal{S}^{(v)} = \{i | \widehat{\Delta}_i < \Delta_i^{(min)}$ or $\widehat{\Delta}_i > \Delta_i^{(max)}\}$;
$\quad$ $\widehat{\Delta}_i = \text{clip}(\widehat{\Delta}_i, \min = \Delta_i^{(min)}, \max = \Delta_i^{(max)}), i \in \mathcal{S}^{(v)}$;
$\quad$ $\mathcal{S}^{(f)} = \mathcal{S}^{(f)} \cup \mathcal{S}^{(v)}$;
$\quad$ Update $\widehat{\Delta}$ according to (5) with elements in $\mathcal{S}^{(f)}$ fixed;
$\quad$ Update $i = i + 1$;
**end**
**Output:** $\|\widehat{\Delta}\|_p$

---

$$
\begin{aligned}
\mathbf{U}_i(\mathbf{x} + \Delta) + \mathbf{p}_i \quad & < 0 \\
\iff \quad \mathbf{U}_i\Delta \quad & < -\mathbf{U}_i\mathbf{x} - \mathbf{p}_i \\
\implies \quad -\|\mathbf{U}_i\|_q\|\Delta\|_p \quad & < -\mathbf{U}_i\mathbf{x} - \mathbf{p}_i \\
\iff \quad \|\Delta\|_p \quad & > \frac{\mathbf{U}_i\mathbf{x} + \mathbf{p}_i}{\|\mathbf{U}_i\|_q}
\end{aligned}
\tag{13}
$$

The $\implies$ comes from the Hölder's inequality. (13) indicates a perturbation of $l_p$ norm over $d_{ic} = \max\left\{0, \frac{\mathbf{U}_i\mathbf{x} + \mathbf{p}_i}{\|\mathbf{U}_i\|_q}\right\}$ is needed to break the sufficient condition of $\mathbf{z}_c'^{(N)} - \mathbf{z}_i'^{(N)} \geq 0$. Based on the assumption of adversarial budget $\mathcal{S}_\epsilon^{(p)}(\mathbf{x})$ when linearizing the model, so the $l_p$ norm of a perturbation to produce an adversarial example is at least $\min\{\epsilon, d_c\}$. $\qquad\square$

## C.2 PROOF OF THE OPTIMALITY OF GEEDY ALGORITHM

*Proof.* We use primal-dual method to solve the optimization problem (4), which is a convex optimization problem with linear constraints.

It is clear that there exists an image inside the allowable pixel space which makes the model predict wrong label. That is to say, the constrained problem (4) is strictly feasible:

$$
\exists \Delta \ s.t. \ \mathbf{a}\Delta + b < 0, \Delta^{(min)} < \Delta < \Delta^{(max)} \tag{14}
$$

Thus, this convex optimization problem satisfies *Slater's Condition* i.e. strong duality holds. We then rewrite the primal problem as:

$$
\begin{aligned}
\min_{\Delta^{(min)} \leq \Delta \leq \Delta^{(max)}} & \|\Delta\|_p^p \\
s.t. \ & \mathbf{a}\Delta + b \leq 0
\end{aligned}
\tag{15}
$$

We minimize $\|\Delta\|_p^p$ instead of directly $\|\Delta\|_p$ in order to decouple all elements in vector $\Delta$. In addition, we consider $\Delta^{(min)} \leq \Delta \leq \Delta^{(max)}$ as the domain of $\Delta$ instead of constraints for simplicity. We write the dual problem of (15) by introducing a coefficient of relaxation $\lambda \in \mathbb{R}_+$.

$$
\max_{\lambda \geq 0} \min_{\Delta^{(min)} \leq \Delta \leq \Delta^{(max)}} g(\Delta, \lambda) \stackrel{\text{def}}{=\!=} \|\Delta\|_p^p + \lambda(\mathbf{a}\Delta + b) \tag{16}
$$

To solve the inner minimization problem, we let the gradient $\frac{\partial g(\Delta, \lambda)}{\partial \Delta_i} = \text{sign}(\Delta_i)p|\Delta_i|^{p-1} + \lambda \mathbf{a}_i$ to be zero and obtain $\Delta_i = -\text{sign}(\mathbf{a}_i)\left|\frac{\lambda \mathbf{a}_i}{p}\right|^{\frac{1}{p-1}}$. Based on the convexity of function $g(\Delta, \lambda)$ w.r.t. $\Delta$, we can obtain the optimal $\tilde{\Delta}_i$ in the domain.

$$\tilde{\Delta}_i = \text{clip}\left(-\text{sign}(\mathbf{a}_i)\left|\frac{\lambda \mathbf{a}_i}{p}\right|^{\frac{1}{p-1}}, \min = \Delta_i^{(min)}, \max = \Delta_i^{(max)}\right) \quad (17)$$

Based on the strong duality, we can say the optimal $\tilde{\Delta}$ is chosen by setting a proper value of $\lambda$. Fortunately, $\|\tilde{\Delta}\|_p$ increases monotonically with $\lambda$, so the smallest $\lambda$ corresponds to the optimum.

As we can see, the expression of $\widehat{\Delta}$ in (5) is consistent with $\tilde{\Delta}_i$ in (17) if $\lambda$ is set properly. [5] Greedy algorithm in Algorithm 2 is the process to gradually increase $\lambda$ to find the smallest value satisfying constraint $\mathbf{a}\Delta + b \leq 0$. With the increase of $\lambda$, elements in vector $\Delta$ remain unchanged when they reach either $\Delta^{(min)}$ or $\Delta^{(max)}$, so we keep such elements fixed and optimize the others.

$\square$

## D  ADDITIONAL EXPERIMENTS

### D.1  DETAILS OF EXPERIMENTS

#### D.1.1  MODEL ARCHITECTURE

FC1 and CNN network used in this paper is identical to ones used in Croce et al. (2018). FC1 network is a fully-connected network with one hidden layer of 1024 neurons. CNN network has two convolutional layers and one additional hidden layer before the output layer. Both convolutional layers have a kernel size of 4, stride of 2 and padding of 1 on both sides, so the height and width of feature maps are halved after each convolutional layer. The first convolutional layer has 32 channels while the second one has 16. The hidden layer following convolutional layers has 100 neurons.

#### D.1.2  HYPER-PARAMETER SETTINGS

In all experiments, we use Adam optimizer (Kingma & Ba, 2014) with the initial learning rate $10^{-3}$ and train all models for 100 epochs with a mini-batch of 100 instances. For CNN models, we decrease the learning rate to $10^{-4}$ for last 10 epochs. When we train CNN models on MNIST, we only calculate the polyhedral envelope of 20 instances subsampled from each mini-batch. When we train CNN models on CIFAR10, this subsampling number is 10. For PER and PER+at, the value of $T$ is always 4. We search in the logarithmic scale for the value of $\gamma$ and in the linear scale for the value of $\alpha$. For the value of $\epsilon$, we ensure its final values are close to the ones used in the adversarial budget $\mathcal{S}_\epsilon^{(p)}(\mathbf{x})$. We compare the constant values with exponential growth scheme for $\epsilon$ but always use constant values for $\alpha$ and $\gamma$. The optimal value we find for different settings are provided as Table 4 below.

### D.2  ADDITIONAL EXPERIMENTAL RESULTS

#### D.2.1  ADDITIONAL RESULTS OF THE MAIN EXPERIMENT

We show additional results including IBP (Gowal et al., 2018) and CROWN-IBP (C-IBP) (Zhang et al., 2019) in Table 5. We also report the certified robust error obtained by IBP (CRE IBP) and the corresponding average certified bound (ACB IBP). Consistent with Zhang et al. (2019), we find IBP has non-trivial certification results only on models that are trained by IBP or CROWN-IBP. For these models, IBP can have even better certification results than Fast-Lin / KW / PEC, whose complexity are significantly higher. As mentioned in Liu et al. (2019), IBP and Fast-Lin are complementary. Our PER and PER+at algorithms still outperform IBP and CROWN-IBP.

---

[5]The power term $\frac{q}{p} = \frac{1}{p-1}$ when $\frac{1}{p} + \frac{1}{q} = 1$

| | $\alpha$ | $\epsilon$ | $\gamma$ |
|---|---|---|---|
| MNIST - FC1, $l_\infty$ | constant, 0.15 | initial value 0.0064 ×2 every 20 epochs | constant, 0.1 |
| MNIST - CNN, $l_\infty$ | constant, 0.15 | constant, 0.1 | constant, 0.3 for PER constant, 0.03 for PER+at |
| CIFAR10 - CNN, $l_\infty$ | constant, 0.1 | constant, 0.008 | constant, $3 \times 10^{-4}$ for PER constant, $1 \times 10^{-3}$ for PER+at |
| MNIST - FC1, $l_2$ | constant, 0.45 | initial value 0.02 ×2 every 20 epochs | constant, 1.0 |
| MNIST - CNN, $l_2$ | constant, 0.45 | constant, 0.3 | constant, 1.0 |
| CIFAR10 - CNN, $l_2$ | constant, 0.15 | constant, 0.1 | constant, 0.3 for PER constant, 1.0 for PER+at |

**Table 4:** Values of $\alpha$, $\epsilon$ and $\gamma$ for different experiments.

| | CTE (%) | PGD (%) | CRE Lin (%) | CRE IBP (%) | ACB KW | ACB IBP | ACB PEC | CTE (%) | PGD (%) | CRE Lin (%) | CRE IBP (%) | ACB KW | ACB IBP | ACB PEC |
|---|---|---|---|---|---|---|---|---|---|---|---|---|---|---|
| | **MNIST - FC1, ReLU, $l_\infty$, $\epsilon = 0.1$** | | | | | | | **MNIST - FC1, ReLU, $l_2$, $\epsilon = 0.3$** | | | | | | |
| plain | 1.99 | 98.37 | 100.00 | 100.00 | 0.0000 | 0.0000 | 0.0000 | 1.99 | 9.81 | 40.97 | 99.30 | 0.1771 | 0.0021 | 0.2300 |
| at | 1.42 | 9.00 | 97.94 | 100.00 | 0.0021 | 0.0000 | 0.0099 | 1.35 | 2.99 | 14.85 | 99.23 | 0.2555 | 0.0023 | 0.2684 |
| KW | 2.26 | 8.59 | 12.91 | 69.20 | 0.0871 | 0.0308 | 0.0928 | 1.23 | 2.70 | 4.91 | 41.55 | 0.2853 | 0.1754 | 0.2892 |
| IBP | 1.65 | 9.67 | 87.27 | 15.20 | 0.0127 | 0.0848 | 0.0705 | 1.36 | 2.90 | 6.87 | 9.01 | 0.2794 | 0.2730 | 0.2876 |
| C-IBP | 1.98 | 9.50 | 67.39 | 14.45 | 0.0326 | 0.0855 | 0.0800 | 1.26 | 2.80 | 6.36 | 8.73 | 0.2809 | 0.2738 | 0.2884 |
| MMR | 2.11 | 17.82 | 33.75 | 99.88 | 0.0663 | 0.0001 | 0.0832 | 2.40 | 5.88 | 7.76 | 99.55 | 0.2767 | 0.0013 | 0.2845 |
| MMR+at | 2.04 | 10.39 | 17.64 | 95.09 | 0.0824 | 0.0049 | 0.0905 | 1.77 | 3.76 | 5.68 | 99.86 | 0.2830 | 0.0004 | 0.2880 |
| PER | 1.60 | 7.45 | 11.71 | 92.89 | 0.0883 | 0.0071 | 0.0935 | 1.26 | 2.44 | 5.35 | 59.17 | 0.2840 | 0.1225 | 0.2888 |
| PER+at | 1.81 | 7.73 | 12.90 | 99.90 | 0.0871 | 0.0001 | 0.0925 | 0.67 | 1.40 | 4.84 | 64.79 | 0.2855 | 0.1056 | 0.2910 |
| | **MNIST - CNN, ReLU, $l_\infty$, $\epsilon = 0.1$** | | | | | | | **MNIST - CNN, ReLU, $l_2$, $\epsilon = 0.3$** | | | | | | |
| plain | 1.28 | 85.75 | 100.00 | 100.00 | 0.0000 | 0.0000 | 0.0000 | 1.28 | 4.93 | 100.00 | 100.00 | 0.0000 | 0.0000 | 0.0000 |
| at | 1.02 | 4.75 | 91.91 | 100.00 | 0.0081 | 0.0000 | 0.0189 | 1.12 | 2.50 | 100.00 | 100.00 | 0.0000 | 0.0000 | 0.0000 |
| KW | 1.21 | 3.03 | 4.44 | 100.00 | 0.0956 | 0.0000 | 0.0971 | 1.11 | 2.05 | 5.84 | 100.00 | 0.2825 | 0.0000 | 0.2861 |
| IBP | 1.51 | 4.43 | 23.89 | 8.13 | 0.0761 | 0.0919 | 0.0872 | 2.37 | 3.85 | 51.12 | 11.73 | 0.1534 | 0.2648 | 0.1669 |
| C-IBP | 1.85 | 4.28 | 10.72 | 6.91 | 0.0893 | 0.0931 | 0.0928 | 2.89 | 4.44 | 31.62 | 12.29 | 0.2051 | 0.2631 | 0.2178 |
| MMR | 1.65 | 6.09 | 11.56 | 100.00 | 0.0884 | 0.0000 | 0.0928 | 2.57 | 5.49 | 10.03 | 100.00 | 0.2699 | 0.0000 | 0.2788 |
| MMR+at | 1.19 | 3.35 | 9.49 | 100.00 | 0.0905 | 0.0000 | 0.0939 | 1.73 | 3.22 | 9.46 | 100.00 | 0.2716 | 0.0000 | 0.2780 |
| PER | 1.44 | 3.44 | 5.13 | 100.00 | 0.0949 | 0.0000 | 0.0965 | 1.02 | 1.87 | 5.04 | 100.00 | 0.2849 | 0.0000 | 0.2882 |
| PER+at | 0.50 | 2.02 | 4.85 | 100.00 | 0.0952 | 0.0000 | 0.0969 | 0.43 | 0.91 | 5.43 | 100.00 | 0.2837 | 0.0000 | 0.2878 |
| | **CIFAR10 - CNN, ReLU, $l_\infty$, $\epsilon = 2/255$** | | | | | | | **CIFAR10 - CNN, ReLU, $l_2$, $\epsilon = 0.1$** | | | | | | |
| plain | 24.62 | 86.29 | 100.00 | 100.00 | 0.0000 | 0.0000 | 0.0000 | 23.29 | 47.39 | 100.00 | 100.00 | 0.0000 | 0.0000 | 0.0000 |
| at | 27.04 | 48.53 | 85.36 | 100.00 | 0.0011 | 0.0000 | 0.0015 | 25.84 | 35.81 | 99.96 | 100.00 | 0.0000 | 0.0000 | 0.0000 |
| KW | 39.27 | 48.16 | 53.81 | 99.98 | 0.0036 | 0.0000 | 0.0040 | 40.24 | 43.87 | 48.98 | 100.00 | 0.0510 | 0.0000 | 0.0533 |
| IBP | 46.74 | 56.38 | 61.81 | 67.58 | 0.0030 | 0.0025 | 0.0034 | 57.90 | 60.03 | 64.78 | 78.13 | 0.0352 | 0.0219 | 0.0366 |
| C-IBP | 58.32 | 63.56 | 66.28 | 69.10 | 0.0026 | 0.0024 | 0.0029 | 71.21 | 72.51 | 76.23 | 80.97 | 0.0238 | 0.0190 | 0.0256 |
| MMR | 34.59 | 57.17 | 69.28 | 100.00 | 0.0024 | 0.0000 | 0.0032 | 40.93 | 50.57 | 57.07 | 100.00 | 0.0429 | 0.0000 | 0.0480 |
| MMR+at | 35.36 | 49.27 | 59.91 | 100.00 | 0.0031 | 0.0000 | 0.0037 | 37.78 | 43.98 | 53.33 | 100.00 | 0.0467 | 0.0000 | 0.0502 |
| PER | 39.21 | 50.98 | 57.45 | 99.98 | 0.0033 | 0.0000 | 0.0038 | 34.10 | 52.54 | 63.42 | 100.00 | 0.0369 | 0.0000 | 0.0465 |
| PER+at | 28.87 | 43.55 | 56.59 | 100.00 | 0.0034 | 0.0000 | 0.0040 | 25.76 | 33.47 | 46.74 | 100.00 | 0.0533 | 0.0000 | 0.0580 |

**Table 5:** Full results of 9 training schemes and 7 evaluation schemes for ReLU networks.

The distribution of certified bounds of CIFAR10 model against $l_\infty$ attack are shown in Figure 6. We only consider the certified bounds strictly larger than 0 but smaller than $\epsilon$, which are non-trivial bounds not obtained by one-shot Fast-Lin / KW. For the CIFAR10 test set, there are $10\% - 20\%$ of such points, and their certified bounds are close to being uniformly distributed between 0 and $\epsilon$.

We show the histogram of parameter values for CIFAR10 models trained by KW, MMR+at and PER+at. It is clear that the parameter values of KW models are much more concentrated around 0 than those of PER+at models. For both $l_\infty$ and $l_2$ attacks, the PER+at model has the biggest norms, evidencing that it better uses the model capacity.

As we can see in Equation 6, our proposed hinge-loss based PER is a constant and does not contribute to the parameter update when $\tilde{d}_{jic}$ is greater than $\alpha$, which is usually $\epsilon$. It means that when a data point is robust enough against the adversarial attack, PER will not push the decision boundary further away from it. However, KW has a cross-entropy based loss. It will always further push the output

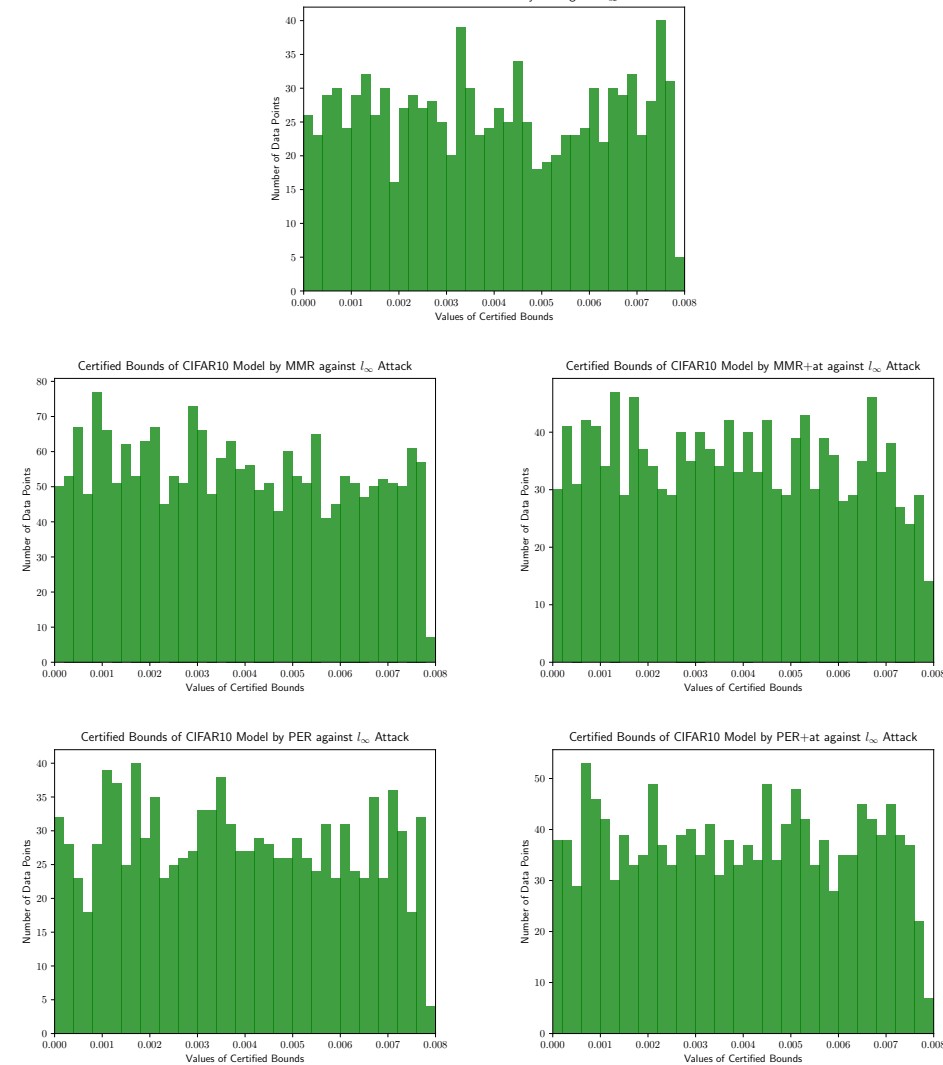

**Figure 5:** Distributions of certified bounds in $(0, \epsilon)$ for CIFAR10 models trained against $l_\infty$ attacks. The total numbers of such test points are 1045, 2162, 1396, 1122, 1489 for KW, MMR, MMR+at, PER, PER+at models, respectively.

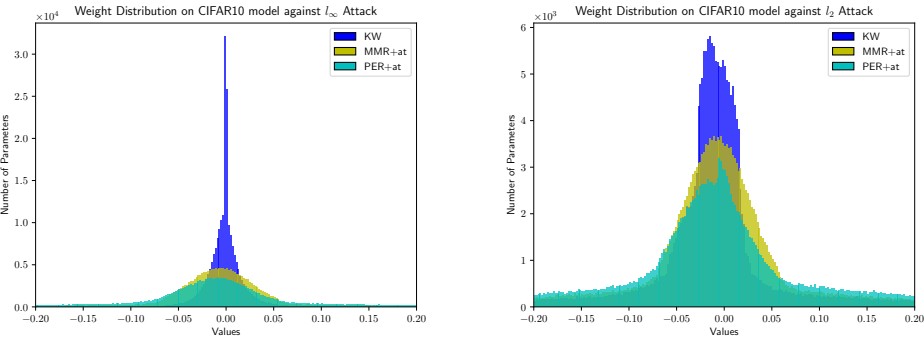

**Figure 6:** Parameter value distributions of CIFAR10 models trained against $l_\infty$ and $l_2$ attacks. The Euclidiean norm of parameter for KW, MMR+at, PER+at model against $l_\infty$ attack is 18.08, 38.36 and 94.63 respectively. For models against $l_2$ attack, the corresponding Euclidiean norm is 71.34, 62.97 and 141.77 respectively.

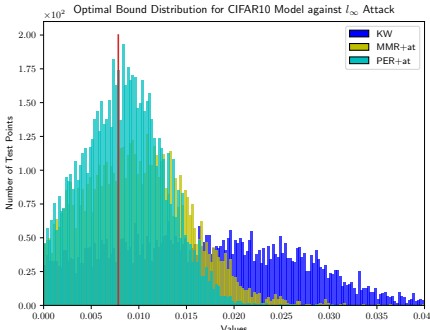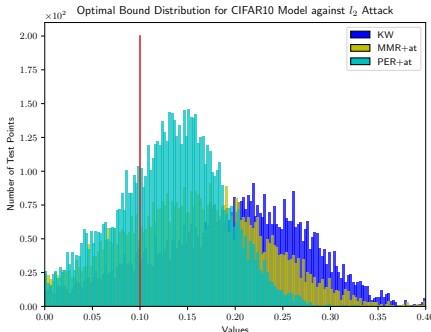

**Figure 7:** The distribution of optimal certified bounds of CIFAR10 models trained against $l_\infty$ and $l_2$ attacks. The target bounds are marked as a red vertical line. ($2/255$ for $l_\infty$ cases and $0.1$ for $l_2$ cases.)

logits of the true label bigger and the others smaller. Figure 7 shows the distribution of optimal $\epsilon$ found by Algorithm 1 for CIFAR10 models against $l_\infty$ and $l_2$ attacks. Table 1 shows that KW and PER+at have similar certified robust error, which is the area on the right of the vertical red line in Figure 7. Because of the property of cross-entropy loss, KW pushes too many points far beyond the robustness target we set. This is unnecessary and causes the loss of clean accuracy.

### D.2.2 TRAINING BY IBP-INSPIRED MODEL LINEARIZATION

We use ReLU networks to compare the performance of IBP-inspired model linearization with the ones used in Fast-Lin / CROWN. We observed that the hyper-parameter settings in Table 4 are ill-suited here. Specifically, since the estimation bounds of the output logits is looser, the gradient of the PER term based on IBP-inspired linearization is larger than in the main experiments. As a result, we used a value $\gamma$ of 3 or 10 times smaller than those in Table 4. Furthermore, starting with a large $\epsilon$ might not work, because the bounds will be even looser under a big adversarial budget. So, we adopt a 'warm up' scheme consisting of increasing the value of $\epsilon$ linearly until it reaches the desired value.

We provide the results in Table 6 below. Note that we still use the tighter linearization method of Fast-Lin / CROWN when we run our certification method (PEC). Despite the looser polyhedral envelope during training, the IBP-inspired model linearization achieves competitive performance with the ones in Fast-Lin / CROWN. In some cases, it has even better performance on the test set. Because of the lower computational complexity and smaller memory consumption, the IBP-inspired method is suitable for machines with limited computational resources.

### D.2.3 SEARCHING FOR THE OPTIMAL VALUE OF $\epsilon$

Tables 7 and 8 show the number of bound calculations in the binary search for the optimal $\epsilon$ in PEC and Fast-Lin / CROWN. In $l_\infty$ cases, the original interval $[\underline{\epsilon}, \bar{\epsilon}]$ is $[0, 0.4]$ for MNIST and $[0, 0.1]$ for CIFAR10. In $l_2$ cases, the original interval $[\underline{\epsilon}, \bar{\epsilon}]$ is $[0, 1.2]$ for MNIST and $[0, 0.4]$ for CIFAR10. The bound of the number calculation does not depend on the model in Fast-Lin / CROWN and is model-dependent in PEC as discussed in Section 4.3.

| | CTE (%) | PGD (%) | Change (%) | CRE (%) | Change (%) | ACB KW | Change | ACB PEC | Change |
|---|---|---|---|---|---|---|---|---|---|
| | **MNIST - FC1, ReLU, $l_\infty$. $\epsilon = 0.1$** | | | | | | | | |
| PER | 1.60 | 6.28 | -1.17 | 11.96 | +0.25 | 0.0880 | -0.0003 | 0.0934 | -0.0001 |
| PER+at | 1.54 | 7.15 | -0.58 | 13.19 | +0.29 | 0.0868 | -0.0003 | 0.0927 | +0.0002 |
| | **MNIST - CNN, ReLU, $l_\infty$. $\epsilon = 0.1$** | | | | | | | | |
| PER | 1.03 | 2.40 | -1.04 | 4.64 | -0.49 | 0.0954 | +0.0005 | 0.0967 | +0.0002 |
| PER+at | 0.48 | 1.29 | -0.73 | 4.61 | -0.24 | 0.0954 | +0.0002 | 0.0971 | +0.0001 |
| | **CIFAR10 - CNN, ReLU, $l_\infty$. $\epsilon = 2/255$** | | | | | | | | |
| PER | 29.34 | 51.54 | +0.56 | 64.34 | +7.11 | 0.0028 | -0.0005 | 0.0036 | -0.0002 |
| PER+at | 26.66 | 43.35 | -0.20 | 57.72 | +1.13 | 0.0033 | -0.0001 | 0.0040 | 0.0000 |
| | **MNIST - FC1, ReLU, $l_2$, $\epsilon = 0.3$** | | | | | | | | |
| PER | 1.21 | 2.59 | +0.15 | 5.34 | -0.01 | 0.2840 | 0.0000 | 0.2888 | 0.0000 |
| PER+at | 0.74 | 1.46 | +0.06 | 7.81 | +2.97 | 0.2766 | -0.0089 | 0.2860 | -0.0050 |
| | **MNIST - CNN, ReLU, $l_2$, $\epsilon = 0.3$** | | | | | | | | |
| PER | 1.11 | 2.16 | +0.29 | 6.37 | +1.33 | 0.2809 | -0.0040 | 0.2851 | -0.0031 |
| PER+at | 0.52 | 1.12 | +0.21 | 7.89 | +2.48 | 0.2763 | -0.0074 | 0.2812 | -0.0066 |
| | **CIFAR10 - CNN, ReLU, $l_2$, $\epsilon = 0.1$** | | | | | | | | |
| PER | 33.94 | 43.06 | -9.48 | 56.80 | -6.62 | 0.0432 | +0.0063 | 0.0484 | +0.0019 |
| PER+at | 24.85 | 31.32 | -2.15 | 47.28 | +0.52 | 0.0528 | -0.0005 | 0.0572 | -0.0008 |

**Table 6:** Results when using IBP-inspired model linearization. The color green, red and gray indicate better, worse and same results compared to those in Table 1

| | $T_{Lin}$ | $T_{PEC}$ | $\frac{T_{PEC}}{T_{Lin}}$ | $T_{Lin}$ | $T_{PEC}$ | $\frac{T_{PEC}}{T_{Lin}}$ | $T_{Lin}$ | $T_{PEC}$ | $\frac{T_{PEC}}{T_{Lin}}$ |
|---|---|---|---|---|---|---|---|---|---|
| | **MNIST - FC1, ReLU, $l_\infty$** | | | **MNIST - CNN, ReLU, $l_\infty$** | | | **CIFAR10 - CNN, ReLU, $l_\infty$** | | |
| plain | | 9.85 | 0.8207 | | 10.56 | 0.8804 | | 9.33 | 0.9331 |
| at | | 10.77 | 0.8972 | | 11.39 | 0.9489 | | 9.12 | 0.9128 |
| KW | | 8.48 | 0.7066 | | 11.61 | 0.9674 | | 8.43 | 0.8432 |
| MMR | 12 | 8.04 | 0.6703 | 12 | 10.68 | 0.8897 | 10 | 8.05 | 0.8053 |
| MMR+at | | 7.68 | 0.6402 | | 11.22 | 0.9351 | | 8.45 | 0.8450 |
| PER | | 9.34 | 0.7780 | | 11.17 | 0.9305 | | 8.61 | 0.8606 |
| PER+at | | 9.38 | 0.7816 | | 11.74 | 0.9784 | | 8.68 | 0.8681 |
| | **MNIST - FC1, ReLU, $l_2$** | | | **MNIST - CNN, ReLU, $l_2$** | | | **CIFAR10 - CNN, ReLU, $l_2$** | | |
| plain | | 9.68 | 0.6914 | | 13.64 | 0.9742 | | 11.73 | 0.9775 |
| at | | 10.44 | 0.7457 | | 13.76 | 0.9829 | | 11.67 | 0.9725 |
| KW | | 7.72 | 0.5514 | | 12.63 | 0.9021 | | 10.23 | 0.8525 |
| MMR | 14 | 5.86 | 0.4186 | 14 | 8.52 | 0.6086 | 12 | 9.05 | 0.7542 |
| MMR+at | | 5.91 | 0.4221 | | 12.13 | 0.8664 | | 10.33 | 0.8608 |
| PER | | 11.47 | 0.8194 | | 13.75 | 0.9819 | | 9.13 | 0.7609 |
| PER+at | | 11.34 | 0.8100 | | 13.72 | 0.9796 | | 10.71 | 0.8926 |

**Table 7:** Number of steps of bound calculation for the optimal $\epsilon$ in Fast-Lin ($T_{Lin}$) and PEC ($T_{PEC}$) for ReLU networks. Note that $T_{Lin}$ is a constant for different models given the original interval $[\underline{\epsilon}, \bar{\epsilon}]$.

| | $T_{CRO}$ | $T_{PEC}$ | $\frac{T_{PEC}}{T_{CRO}}$ | $T_{CRO}$ | $T_{PEC}$ | $\frac{T_{PEC}}{T_{CRO}}$ | $T_{CRO}$ | $T_{PEC}$ | $\frac{T_{PEC}}{T_{CRO}}$ |
|---|---|---|---|---|---|---|---|---|---|
| | **MNIST - FC1, Sigmoid, $l_\infty$** | | | **MNIST - FC1, Tanh, $l_\infty$** | | | **MNIST - FC1, ArcTan, $l_\infty$** | | |
| plain | | 9.06 | 0.7547 | | 9.69 | 0.8078 | | 8.76 | 0.7304 |
| at | 12 | 10.92 | 0.9104 | 12 | 11.23 | 0.9356 | 12 | 11.19 | 0.9326 |
| PER | | 8.84 | 0.7369 | | 9.02 | 0.7513 | | 8.84 | 0.7369 |
| PER+at | | 9.67 | 0.8060 | | 9.64 | 0.8030 | | 9.62 | 0.8020 |
| | **MNIST - FC1, Sigmoid, $l_2$** | | | **MNIST - FC1, Tanh, $l_2$** | | | **MNIST - FC1, ArcTan, $l_2$** | | |
| plain | | 10.55 | 0.7534 | | 10.96 | 0.7831 | | 9.99 | 0.7134 |
| at | 14 | 11.83 | 0.8447 | 14 | 12.64 | 0.9032 | 14 | 12.30 | 0.8783 |
| PER | | 12.09 | 0.8633 | | 12.20 | 0.8717 | | 12.08 | 0.8626 |
| PER+at | | 12.22 | 0.8728 | | 12.50 | 0.8928 | | 12.30 | 0.8786 |

**Table 8:** Number of steps of bound calculation for the optimal $\epsilon$ in CROWN ($T_{CRO}$) and PEC ($T_{PEC}$) for non-ReLU networks. Note that $T_{CRO}$ is a constant for different models given the original interval $[\underline{\epsilon}, \bar{\epsilon}]$.

