# OpenReview forum: "Training Provably Robust Models by Polyhedral Envelope Regularization"
_ICLR.cc/2020/Conference — Reject_

### Official Review · AnonReviewer3 · 2019-10-23
**Official Blind Review #3**

**Rating:** 8

**Review:**

Summary:
There exists several method (KW, Crown, zonotope transformers) which essentially propagate linear lower and upper bound through the network (bounds is a linear function of the input variables). So far, to check for robustness, these methods propagated the bounds until the last layer (+ margin computation) and then concretized them and compared the result to zero to see if the studied radius was safe or not. This paper highlights the fact that in the case where the studied radius is not safe, it is possible to extract a safe radius from the linear bound.

The authors then show that:
-> this can be used to make binary search to find the larger verifiable epsilon faster (by providing a lower bound on epsilon even when failing to verify, rather than simply when succeeding verification)
-> this can be employed during the training process to improve regularization, similarly to another previously proposed method (Croce et al.). While the methods are similar, the bounds are much less conservative than Croce's one and should therefore be more helpful

The presentation of the content is quite clear, and Figure 1.a is extremely useful in conveying the benefits of the method.

Experimental validation is thorough:
-> Comparing the ACB KW and ACB PEC columns in Table 1 and 2, for all type of activation functions, shows that this is as expected a strict improvement in the generated bound.
-> Comparing the various rows in Table 1 shows the improvement with regards to other methods, both in terms of clean accuracy (this produces less over-regularization), while maintaining similar or better robust accuracy.
-> Table 6 shows the benefit in the application to binary search.

Opinion:
This paper is clearly written and I think that it's an interesting insight and that the authors do a good job at conveying its usefulness. I'm happy for this paper to be accepted.



**Experience Assessment:**

I have published in this field for several years.

**Review Assessment: Checking Correctness Of Derivations And Theory:**

I carefully checked the derivations and theory.

**Review Assessment: Checking Correctness Of Experiments:**

I carefully checked the experiments.

**Review Assessment: Thoroughness In Paper Reading:**

I read the paper thoroughly.

---

> ### Author Response · Authors · 2019-11-12
> **Thank You**
>
> We thank the reviewer very much for their positive comments. Following the other reviewers’ suggestions, we have added more experiments as stated in the revision summary.

---

### Official Review · AnonReviewer2 · 2019-10-23
**Official Blind Review #2**

**Rating:** 3

**Review:**

This paper proposes a certifiable NN training method, "polyhedral envelope
regularization" (PER) for defending against adversarial examples.  The defense
is based on the same linear relaxation based outer bounds of neural networks
(KW/CROWN) used in many previous works.  The paper makes a few new (but small)
technical contributions:

1. this paper uses a different loss function (7), which is essentially Hinge
loss on the lower bounds of distance to decision boundary. Previous works like
KW used cross-entropy loss on the lower bound of prediction margin instead,
which was based on minimax robust optimization theory. But I am not fully
convinced if the new loss function is better or not.

2. in (5), the authors solve the bounded input case more carefully than
previous works.  (5) is trivial to solve in the L infinity case and has been
used in previous works like (Wong & Kolter 2018, Gowal et al., 2018 and Zhang
et al., 2019); but solving it for other norms requires some efforts, and this
paper proposes a good solution for it (Algorithm 2);

3. In previous works like KW/CROWN, to find the largest certifiable radius, a
binary search is needed. The authors proposes a very small improvement to the
binary search process by setting the lower bound of search to the largest
epsilon that is certifiable using the current linear relaxations obtained from
a larger epsilon.

The authors does not improve any bounds proposed in KW/CROWN, and they reuse
the same bounds. I see the main contribution as the new hinge-like loss
function for training, and a more careful procedure to find the largest
certifiable radius in bounded input case.

Empirically, the improvement of the proposed algorithm is limited - based on
Table 1 it is hard to say if PER is better than KW or not. PER+at outperforms
KW sometimes, however it is not a completely fair comparison, as we can add a
PGD based adversarial training loss to KW as well, as done in DiffAI (Mirman et
al., https://github.com/eth-sri/diffai).

Questions:

1. In my personal experience I usually found Hinge loss not as effective as
cross-entropy loss in deep learning based tasks probably due to its
non-smoothness. The claim that (7) is better than cross-entropy loss is that it
does not overregularize the network. The authors should provide more evidence
to show if this argument holds, e.g., plotting the norm of weight matrices
during the training for the two losses to show that it can reduce
overregularization.

2. I think the metric ACB KW and ACB CRO (average certified radius of KW/CROWN)
in Table 1 and 2 are confusing and not fair. In KW and CROWN's evaluation,
giving an epsilon, if an example cannot be certified due to epsilon to large
(i.e., ||A|| \epsilon + b > 0), certifiable radius will be count as 0 (flat
line in Figure 1(a)). In this paper, the authors instead in this case use -b /
||A|| as the certifiable radius. This is merely a different way of evaluation,
and I don't see this as a contribution, as the "improvement" does not come from
a tighter bound.  In the same sense, I don't think Figure 1(a) and the
discussions on page 3 are appropriate characterization of KW/CROWN. PEC uses
exactly the same linear bounds as in KW/CROWN, and has the same certification
power.

3. For L2 based perturbations, in Table 1, the epsilon used for MNIST is too
small. It is better to use an epsilon that is aligned with previous works. For
example in Wong et al., 2018 (https://arxiv.org/pdf/1805.12514.pdf), page 22,
you will find the epsilon used for MNIST and CIFAR.

4. As discussed above, it is probably not fair to compare PER+at with KW. A new
baseline like KW+at should also be considered.

5. For norms other than L infinity norms, solving (5) for getting $d$ can be
time consuming (Algorithm 2). How much additional time does it comparing
to KW?

Overall, I cannot recommend accepting this paper due to its limited theoretical
contribution as well as unconvincing empirical results comparing to previous
methods. I suggest rephrasing some parts of the paper and providing more
experimental results as discussed above.


**Experience Assessment:**

I have published in this field for several years.

**Review Assessment: Checking Correctness Of Derivations And Theory:**

I carefully checked the derivations and theory.

**Review Assessment: Checking Correctness Of Experiments:**

I carefully checked the experiments.

**Review Assessment: Thoroughness In Paper Reading:**

I read the paper thoroughly.

---

> ### Author Response · Authors · 2019-11-12
> **Detailed Response to Reviewer #2**
>
> We thank the reviewer for the constructive comments and answer their questions below.
>
> 1. In the Figure 6 of the revised paper, we provide the distribution of the parameter values for KW/MMR+at/PER+at models on CIFAR10. We can see that the models trained by PER+at have larger parameter values than KW. The over-regularization issue of KW was also mentioned in [Zhang 2019, Dvijotham UAI19]. Furthermore, note that models trained using MMR+at, which also introduces a hinge-loss like regularization, are also less over-regularized than those trained using KW.
>
> 2. Tables 1, and 2 report certified bounds obtained in ONE evaluation. That is, in these tables, we compare ONE-SHOT KW/Fast-Lin/PEC. By using binary search, KW/Fast-Lin can obtain the same optimal $\epsilon$ as PEC. As stated in Section 3.1, we do not claim a tighter bound but a finer-grained one, which contributes to faster convergence when searching for the optimal $\epsilon$. Apart from CROWN-style linearization, PEC can be applied to any other method that gives linear bounds of model's output, such as the IBP-inspired one used in Appendix A.3. Furthermore, we would like to clarify that our main contribution, as our title indicates, is to incorporate a geometry-inspired bound to train provably robust models. Our experiments also focus on how PER/PER+at train robust models better than others techniques.
>
> 3. For the results in Table 1, we used the same settings as in [Croce 2019], so that we could directly use the checkpoints provided by them as baselines. We also test the case used in [Wong 2018] where $\epsilon$ for the $l_2$ attack in MNIST is 1.58. The robust error for the MNIST-CNN model (exactly the same as ‘Small model’ in [Wong 2018]) trained by PER is 55.10%, which is better than what [Wong 2018] reports (56.48% in Table 4, Page 22). The error on clean test data for the PER-trained model is 9.00% and lower than the 11.86% reported in [Wong 2018]. We will release all the checkpoints when our code is made public.
>
>
> 4. We have carefully checked the DiffAI paper [Mirman ICML18, Mirman 2019] and do not think that it is a combination of KW and PGD. DiffAI is built on the abstract transformer for Zonotope, and their bound is looser than the KW one, based on Definitions 4.1 - 4.4 in [Mirman ICML18]. DiffAI trains models with the cross-entropy loss using over-approximated bounds of the output logits within the adversarial budget. This is the 'Adversarial Training' mentioned in Section 5 of [Mirman ICML18], which differs from PGD [Madry ICLR18]. [Mirman ICML18] only uses the PGD error as a lower bound of the true robust error and does not incorporate PGD into their algorithm. Actually, in both MMR [Croce 2019] and PER, adversarial training (at) is only used when we calculate the distance between a data point and the estimated decision boundary, because intuitively an adversarial example is closer to the decision boundary. Note that our polyhedral envelope, which is based on a linear approximation, is always calculated on the clean input, because the adversarial budget is defined based on the clean input (second to last paragraph of Section 3.2). By contrast, in KW, the loss is directly calculated based on the cross-entropy of the output logits’ bounds, which can only be estimated based on the clean input. We do therefore not see how KW can be combined with PGD adversarial training.
>
> 5. As discussed in Section 5, for an N-layer network with k, m, n as the dimensions of the output, the input and the hidden layers ($n >> k, m$), the complexity of a CROWN-style linearized bound calculation is $O(N^2 n^3)$. However, based on Equation (18) for both the $l_\infty$ and $l_2$ norm, the complexity of each iteration in Algorithm 2 is $O(km)$, which is significantly smaller than $O(N^2n^3)$. In practice, the loop in the algorithm stops within 5 iterations in almost all cases (Appendix B.2), so the overhead is small. We have run 10 times  PEC and CROWN on the CIFAR10-CNN model. To process the entire test set on a single-GPU machine, the mean and standard deviation in $l_\infty$ cases are 217.51(1.95) for CROWN and 219.16(3.23) for PEC; in $l_2$ cases, they are 236.95(1.64) for CROWN and 239.41(1.92) for PEC. The difference is negligible and within the variance of the 10 attempts.

---

> > ### Comment · AnonReviewer2 · 2019-11-15
> > **Thanks for the response and additional results.**
> >
> > 2. The contribution of KW/CROWN is to provide linear upper and lower bounds for neural network. The bounds proposed by this paper is based on them and have exactly the same power. The way to find epsilon is only a very minor difference in evaluation. I don't think the claim that the proposed"geometry-inspired" algorithm beats KW/CROWN is acceptable, as the "improvement" does not come from a tighter bound. The "geometry-inspired bound" looks too trivial to me, and I believe it is better to not claim it as a main contribution.
> >
> > 4. For adversarial training, you can just simply add a PGD loss term for KW. The PGD loss is not related to the bounds. You just ran PGD attack given the current example $x$ and use the point $x^\prime$ after attack for the regular cross-entropy loss. The same technique can also be applied to IBP. In my experience IBP+PGD loss can improve IBP by a few percent. Without adversarial training, the proposed method (PEC) cannot outperform other methods. So I am not sure if the improvement shown in this paper (the best method is PEC+at) is from adversarial training or PEC itself.
> >
> > Unfortunately, due to the main concerns above, I cannot increase my rating.

---

> > > ### Author Response · Authors · 2019-11-15
> > > **Additional Response to Reviewer #2**
> > >
> > > We thank the reviewer for their response.
> > >
> > > Regarding comment 2, we agree with the reviewer that this is not our main contribution. We have updated the last paragraph of the introduction to highlight that our main contribution is to propose a regularization scheme to train provably robust networks without over-regularizing the model. The corresponding certification part, i.e., PEC, is one important step to build PER regularization. We indeed clarify that PEC can be based on any linearization method and we use exactly the same linear bound as Fast-Lin in our main experiment. The advantage of PEC over Fast-Lin/KW is that it has finer-grained one-shot certified bounds, which contributes to faster convergence to the optimal $\epsilon$ (Table 6 & 7). This is a by-product of geometric-inspired certification and we put the results in the Appendix. In the main text, we focus on comparing different training methods.
> > >
> > > Regarding question 4, we have tested combining PGD adversarial training with KW loss as you proposed. Let’s define the hybrid loss $L = L_{PGD} + \eta L_{KW}$ where $L_{PGD}$ is the loss of PGD adversarial training and $L_{KW}$ is the one of KW in [Wong 2018]. Note that both parts are cross-entropy losses. The PGD loss and KW loss are the lower bound and upper bound, respectively, of the true worst case loss under the adversarial budget. We have tested the case when coeficient $\eta = 1$ and $\eta = 10.$ on the MNIST dataset with the same experimental setting as the ones in Table 1. We use Adam optimizer, so the results are invariant to the scaling of training objective functions. The results are as follows (the notation is the same as Table 1):
> > >
> > > MNIST-FC1, ReLU, $l_\infty$, $\epsilon = 0.1$, $\eta = 1.$
> > > CTE =  0.81%, PGD = 7.44%, CRE Lin = 23.29%, ACB KW = 0.0767, ACB PEC = 0.0890.
> > > MNIST-FC1, ReLU, $l_\infty$, $\epsilon = 0.1$, $\eta = 10.$
> > > CTE = 1.60%, PGD = 7.99%, CRE Lin = 13.74%, ACB KW = 0.0863, ACB PEC = 0.0928.
> > > MNIST-CNN, ReLU, $l_\infty$, $\epsilon = 0.1$, $\eta = 1.$
> > > CTE = 0.81%, PGD = 3.64%, CRE Lin = 8.12%, ACB KW = 0.0919, ACB PEC = 0.0952.
> > > MNIST-CNN, ReLU, $l_\infty$, $\epsilon = 0.1$, $\eta = 10.$
> > > CTE = 1.96%, PGD = 4.88%, CRE Lin = 7.21%, ACB KW = 0.0928, ACB PEC = 0.0952.
> > >
> > > We notice that the results are worse than KW regarding the certified robust error (CRE Lin). The performance is between the PGD-trained model and KW-trained model. For the cases when $\eta = 1$, the obtained model has better empirical clean accuracy and PGD robust accuracy. For a bigger $\eta$, the loss is more similar to KW, i.e., the model has better certified accuracy at the cost of clean accuracy.
> > >
> > > Regarding the IBP training cases you mentioned, we need to point out that the original IBP paper [Gowal 2018] has a vanilla cross-entropy term. It is because IBP uses very aggressive approximation and the vanilla cross-entropy term is needed to stabilize the training. However, the original KW paper [Wong ICML18] does not use this term. In addition, we did not see any existing work combining KW with PGD as what you proposed.
> > >
> > > To argue the advantages of hinge-loss based PER over cross-entropy-loss based KW, we provide the following explanation and supporting evidence: when a data point is robust enough against adversarial attack (in our case, that means its certified bound is bigger than $\epsilon$), our hinge-loss based PER is a constant and does not contribute to the update of the model parameters. However, cross-entropy-loss based KW will always further push the output logits of the true labels higher and the false ones lower. This leads to the problem of over-regularization of KW-trained models.  If we search for the optimal bounds by Algorithm 1 for each test point in the CIFAR10 $l_\infty$ case, we find that the average optimal bounds for KW-trained model is 0.0169 while for PER+at model it is only 0.0084.  While KW and PER+at have similar certified error on the target $\epsilon = 2 / 255 = 0.0078$, Figure 7 of Appendix D.2.1 shows that KW pushes too many points far beyond the robustness target we set. This is unnecessary and at the cost of clean accuracy.

---

### Official Review · AnonReviewer4 · 2019-10-27
**Official Blind Review #4**

**Rating:** 3

**Review:**

The paper proposes an approach for computing more refined estimates of robustness in comparison w/ existing linear approximation approaches that only give a yes or no answer with regard to robustness guarantees for a given lp-norm ball with radius epsilon. The nice thing is that as the linear-approximations get better, the contributions in this paper would continue to help.

The paper makes two key algorithmic/theoretical contributions:
1. An approach to obtain a better estimate of the radius of the l-p ball where the NN is provably robust. This result is fairly straightforward, and relies on computing the distance of a point to the boundary of adversarial polytope.

2. An approach to exploit the fact that the pixel values are restricted to specific bounds, which might allow us trim away some regions from the l-p norm balls around a given input image w.r.t which we want to be robust, while computing the robustness. This I think is a more interesting contribution.


I am leaning towards a reject, however I am open to changing my score. I have several key concerns:

1. Verified Training: Why is there no comparison with IBP and IBP+Crown (Zhang 2019) -- it seems like an appropriate comparison to make. Particularly, when the current paper refers and discusses both of the above works.

2. I am not sure that comparison with CRO entirely suffices in my opinion. Would it be possible to compare with the tighter SDP based approaches (Raghunathan et al., NeurIPS'19 and Dvijotham et al., UAI'19)? Is there a specific reason to not compare (other than that the SDP based approach is not a linear approximation, and probably is much slower)?

My main concern here is the utility of pushing boundaries with the linear approximation, while there are potentially tighter relaxations?

3. You claim no overhead compared to CROWN. Don't the greedy-optimization steps add some overhead, or am I missing something? How expensive are they? (It's possible I might have missed some discussion in this regard. If so, please point me in the right direction and that should suffice)

4. Can you plot the distributions of the certified epsilon? Are there a few samples for which you can certify a much larger epsilon (than just saying not robust) or are there a lot of samples where you can only show a tiny bit of robustness (compared to CROWN saying not robust)?

The gains in the average robustness are somewhat small, and these gains alone are not convincing without being able to see how these gains were obtained.

Minor Comment:
Missing reference to MixTrain for B' < B helps.

Update:
Thanks for the detailed response!

1. This makes sense, the first bit seems obvious --> if you don't train with IBP, you won't get much out of IBP and this is reasonably well known.

Table 5: When trained with IBP and verifying with IBP, it seems to do better or quite comparable to train/verify with PER --- in this sense, the gains seem quite marginal.

2. Since a part of the contribution claimed in this paper is improved robustness guarantees for pre-trained networks, I do feel that comparisons with the UAI'19 paper or the NeurIPS'18 paper would be nice -- however, I do agree that the computational tractability of KW/FastLin/IBP are much more favorable.

3. Thanks -- it is much more clear now.

4. The distributions for MMR/PER seem quite similar and most of the gain seems to come at the lower end (small eps).
This still remains my biggest concern -- I was hoping that the distributions would diverge a bit more at larger eps, but this is difficult to confirm with the current set of plots.

These plots, as of the current version, are not very useful -- a CDF plot as opposed to a histogram would be much more illustrative in terms of comparing the different approaches. Overlaying them with different opacity might also be useful.

I am still leaning towards a weak reject. I am not sure how the scoring system works, the scores are distributed unevenly -- I would judge this a 5 on the 1-10 scale. However, going by the wording, I will stay with a 3.

**Experience Assessment:**

I have published one or two papers in this area.

**Review Assessment: Checking Correctness Of Derivations And Theory:**

I assessed the sensibility of the derivations and theory.

**Review Assessment: Checking Correctness Of Experiments:**

I assessed the sensibility of the experiments.

**Review Assessment: Thoroughness In Paper Reading:**

I read the paper at least twice and used my best judgement in assessing the paper.

---

> ### Author Response · Authors · 2019-11-12
> **Detailed Response to Reviewer #4**
>
> We thank the reviewer for the constructive comments and address their concerns below.
>
> 1. We have added the results of IBP and CROWN-IBP in Table 5. IBP differs from KW/Fast-Lin/CROWN in that it is not based on the linearization of the activation functions. As in  [Zhang 2019], we observed that IBP is effective at certifying models trained by IBP or CROWN-IBP. In these cases, the more complicated KW/Fast-Lin/CROWN can yield suboptimal results. For other models, IBP is always much worse than KW/Fast-Lin/CROWN. Theoretically, IBP and Fast-Lin are complementary. Our PER and PER+at algorithms still outperform IBP and CROWN-IBP.
>
> 2. Note that both [Raghunathan NeurIPS18] and [Dvijotham UAI19] you mentioned are pure certification method, and thus do not introduce any algorithms to train provably robust models. Despite providing a tighter bound than methods based on linearizing activation functions, the technique of [Dvijotham UAI19], which is a more scalable version of [Rahunathan NeurIPS18], is much slower than the ones in our paper. [Dvijotham UAI19] takes 130 seconds on average (Section 5.2) to certify one point in MNIST for a fully-connected network, while our method requires less than 0.01 second per point on average on MNIST for LeNet, which has much more neurons than the models used in [Dvijotham UAI19]. The difference mainly arises from the number of optimization steps in Algorithm 1 in [Dvijotham UAI19], which varies from a few hundred to tens of thousands (Section 4.4 in Dvijotham UAI19). In other words, there are indeed tighter bounds than linear approximations, such as SDP-based ones you mentioned and all complete certifiers in which no approximations are used. However, these methods all have prohibitively high complexity and cannot be incorporated into training. Our experimental results in Table 1 and 2 show the effectiveness of pushing the decision boundary based on a linear approximation to obtain robust models.
>
> 3. As discussed in Section 5, for an N-layer network with k, m, n as the dimensions of the output, the input and the hidden layers ($n >> k, m$), the complexity of a CROWN-style linearized bound calculation is $O(N^2 n^3)$. However, based on Equation (18) for both the $l_\infty$ and $l_2$ norm, the complexity of each iteration in Algorithm 2 is $O(km)$, which is significantly smaller than $O(N^2n^3)$. In practice, the loop in the algorithm stops within 5 iterations in almost all cases (Appendix B.2), so the overhead is small. We have run 10 times  PEC and CROWN on the CIFAR10-CNN model. To process the entire test set on a single-GPU machine, the mean and standard deviation in $l_\infty$ cases are 217.51(1.95) for CROWN and 219.16(3.23) for PEC; in $l_2$ cases, they are 236.95(1.64) for CROWN and 239.41(1.92) for PEC. The difference is negligible and within the variance of the 10 attempts.
>
> 4. We have provided the distribution of certified epsilon in CIFAR10 as a histogram in Figure 5 of the revised paper. For example, for models trained against $l_\infty$ attack by PER+at, 1489 points in the test set (14.89%) are not certified by one-shot KW, but for which PEC gives a non-zero certified bound. The distribution of these bounds is close to uniform between 0 and $\epsilon$.

---

### Author Response · Authors · 2019-11-12
**Revision Summary**

We have made the following revisions:

1. We have added IBP [Gowal 18] and CROWN-IBP [Zhang 19] as new methods to compare in the main experiments. We have also added IBP as a certification method. The full results of 9 training and 7 evaluation schemes are demonstrated in Table 5 of Appendix D.2.1. Our proposed PER/PER+at method still outperform IBP / CROWN-IBP.
2. We have provided the distribution of the certified bounds by one-shot PEC (Figure 5).
3. To better demonstrate that our method produces less over-regularized models than the baselines, we have added the histogram of their parameter values and report their norms (Figure 6). PER+at models have bigger norms than baselines.
4. We have provided the distribution of optimal $\epsilon$ found by Algorithm 1 for CIFAR10 models  (Figure 7). Given the robustness target, KW pushes too many test points far beyond this target. This is unnecessary and causes the loss of clean accuracy. Our proposed PER+at does not have such over-regularization problems.

---

### Decision · Program_Chairs · 2019-12-19

**Decision:**

Reject

**Comment:**

The authors develop a new technique for training neural networks to be provably robust to adversarial attacks. The technique relies on constructing a polyhedral envelope on the feasible set of activations and using this to derive a lower bound on the maximum certified radius. By training with this as a regularizer, the authors are able to train neural networks that achieve strong provable robustness to adversarial attacks.

The paper makes a number of interesting contributions that the reviewers appreciated. However, two of the reviewers had some concerns with the significance of the contributions made:
1) The contributions of the paper are not clearly defined relative to prior work on bound propagation (Fast-Lin/KW/CROWN). In particular, the authors simply use the linear approximation derived in these prior works to obtain a bound on the radius to be certified. The authors claim faster convergence based on this, but this does not seem like a very significant contribution.

2) The improvements on the state of the art are marginal.

These were discussed in detail during the rebuttal phase and the two reviewers with concerns about the paper decided to maintain their score after reading the rebuttals, as the fundamental issues above were not

Given these concerns, I believe this paper is borderline - it has some interesting contributions, but the overall novelty on the technical side and strength of empirical results is not very high.